

# A methodology for psycho-biological assessment of stress in software engineering

Jan-Peter Ostberg[1], Daniel Graziotin[1], Stefan Wagner[1] and Birgit Derntl[2]

[1] Institute of Software Engineering, University of Stuttgart, Stuttgart, Germany
[2] Department of Psychiatry and Psychotherapy, University of Tübingen, Tübingen, Germany

## ABSTRACT

Stress pervades our everyday life to the point of being considered the scourge of the modern industrial world. The effects of stress on knowledge workers causes, in short term, performance fluctuations, decline of concentration, bad sensorimotor coordination, and an increased error rate, while long term exposure to stress leads to issues such as dissatisfaction, resignation, depression and general psychosomatic ailment and disease. Software developers are known to be stressed workers. Stress has been suggested to have detrimental effects on team morale and motivation, communication and cooperation-dependent work, software quality, maintainability, and requirements management. There is a need to effectively assess, monitor, and reduce stress for software developers. While there is substantial psycho-social and medical research on stress and its measurement, we notice that the transfer of these methods and practices to software engineering has not been fully made. For this reason, we engage in an interdisciplinary endeavor between researchers in software engineering and medical and social sciences towards a better understanding of stress effects while developing software. This article offers two main contributions. First, we provide an overview of supported theories of stress and the many ways to assess stress in individuals. Second, we propose a robust methodology to detect and measure stress in controlled experiments that is tailored to software engineering research. We also evaluate the methodology by implementing it on an experiment, which we first pilot and then replicate in its enhanced form, and report on the results with lessons learned. With this work, we hope to stimulate research on stress in software engineering and inspire future research that is backed up by supported theories and employs psychometrically validated measures.

## INTRODUCTION

Our modern industrialized world is moving fast and demands a lot from the workers within its system, which leaves them stressed out. Some consider stress the scourge of the modern industrial world (*De Jonge et al., 1998*). Stress is a response to exceeding demands placed upon the body or mind (*Selye, 1976*). It is well known that stress is highly related to the deterioration of physical and mental health (*Sonnentag et al., 1994*;

Corresponding author
Jan-Peter Ostberg,
jan-peter.ostberg@informatik.uni-stuttgart.de

*Kaufmann, Pornschlegel & Udris, 1982*). Individuals who perceive a large amount of stress have an increased risk of premature death, coronary heart disease, and mental disorders such as depression or burn-out, as the World Health Organization realized as early as *World Health Organization and Others (1969)* and continued to investigate the problem (*World Health Organization, 2005*). Stress as well as the mere anticipation of stress (*Hyun, Sliwinski & Smyth, 2018*) also has a negative impact on the quality of products, as it increases workers error rates (*Akula & Cusick, 2008*). Workplace environments that are characterized by physical work have been improved over time by research on ergonomic tools as well as their placement to lessen physical strain, and alternation of tasks and restructuring of processes to counter dulling repetitive work. The aim is the replenishment of physical and cognitive resources which were consumed by the stressful tasks. Still, the stress experienced by knowledge workers, like software developers, has a wide range of research opportunities in terms of understanding and preventing generation of (chronic) stress and its effects (*Meier et al., 2018*; *Ostberg et al., 2017*).

Software developers are stressed workers. Short time to market, requirements originating from legislators leading to high penalty payments, fast-changing technological environments (*Chilton, Hardgrave & Armstrong, 2010*), the need to plan for software legacy and obsolescence and interaction with customers (*Rajeswari & Anantharaman, 2003*), as well as possible time zone, linguistic, and cultural differences (*Amin et al., 2011*) are just the tip of the iceberg for potential long-term stressors in software development. Day to day stressors such as cryptic error messages, unintuitive integrated development environments (IDEs), changing requirements which cause high cognitive stain, should be kept as low as possible to avoid an additional burden to body and mind (*Graziotin et al., 2017*).

Stress has been suggested to have detrimental effects on defect rates, team morale as well as motivation, software quality, maintainability, and requirements management (*Meier et al., 2018*). While short-term stress can be pushing and beneficial to software engineers, preliminary research suggests that we should find ways to reduce stress and develop tools for software development that help to reduce stress or at least are no sources of stress (*Brown et al., 2018*). We discussed ways to reduce stress of developers elsewhere (*Ostberg et al., 2017*), but without strong and validated, yet easy to adopt methodologies to detect, assess, and understand stress responses of individuals and groups of developers, it is hard to produce sound statements on the efficiency of any stress reduction approach. For detecting stress, research in software engineering has focused on machine learning and data mining approaches, wearable technologies (*Suni Lopez, Condori-Fernández & Catala Bolos, 2018*), and ad-hoc questionnaires (*Meier et al., 2018*) so far *Brown et al. (2018)* have offered a review of the few scattered studies—but we still see some research gaps, which we highlight in the next section, in terms of discovered knowledge as well as the way we borrow from established research from other disciplines.

Hence, we are motivated to engage in an interdisciplinary endeavor between researchers in software engineering as well as medical and social science fields towards a better understanding of stress while developing software.

This article offers two main contributions. First, we provide an overview of supported theories of stress and the many ways to assess stress in individuals. Second, we propose a robust methodology to detect and measure stress in controlled experiments that is tailored to software engineering research. The methodology has been supported by two controlled experiments which we report on together with lessons learned. With this work, we hope to stimulate research on stress in software engineering and inspire future research that is backed up by supported theories and employs psychometrically validated measurement instruments.

## RELATED WORK

There is substantial psycho-social and medical research on stress and its measurement (*Brown et al., 2018*) but the transfer to software engineering has yet to be made. This is also due to the many medical, psychological, and biological ways to measure stress and on how to report the results (*Noack et al., 2019*) creating the need for interdisciplinary work which increases the complexity of research projects. Furthermore, we believe that solid theoretical and methodological foundations should be a first step towards a better understanding of stress reactions of developers, as it should be with any psychological construct.

Software engineering research, regretfully, is a long way from adopting rigorous and validated research artifacts. *Feldt et al. (2008)* argued in favor of systematic studies of human aspects of software engineering, more specifically to adopt measurement instruments that come from psychology. Seven years after the statement by *Feldt et al. (2008)* and *Graziotin, Wang & Abrahamsson (2015a)* explained that research on the emotional responses of software developers has been threatened by a lack of understanding the underlying constructs, in particular to exchange affect-related psychological constructs such as emotions and moods with motivation, commitment and well-being. The article offers a clarification of these constructs and presents validated measures. Meanwhile, *Lenberg, Feldt & Wallgren (2015)* had published a systematic literature review of studies of human aspects that made use of behavioral science. They called the field behavioral software engineering and found when conducting this kind of research that there are still several knowledge gaps and that there have been very few collaborations between software engineering and social science researchers. *Graziotin, Wang & Abrahamsson (2015b)* have also extended their prior observations to a broader view of software engineering research. Given, that much research in the field has misinterpreted, if not ignored, validated methodology and measurement instruments coming from psychology, the work offered brief guidelines to select a theoretical framework and validated measurement instruments from psychology. This includes a thorough evaluation of the psychometric properties of candidate instruments, which was later echoed in guidelines by *Gren (2018)* and *Wagner et al. (2020)*. *Wagner et al. (2020)* have highlighted a major case of such misconduct, which is evident from the systematic literature review of personality research in software engineering by *Cruz, Da Silva & Capretz (2015)*. The study review found that 48% of personality studies in software engineering use the Myers-Brigg Type Indicator (MBTI), which was known more than 20 years earlier to provide close to no validity and

reliability properties (*Pittenger, 1993*), meaning that the results of about half studies of personality in software engineering research are unlikely reflecting on personality in their conclusions.

The software engineering body of knowledge on stress is quite small and also lacking much understanding of the phenomenon (*Amin et al., 2011*; *Brown et al., 2018*), even though there are few scattered studies that we can review.

Prolonged exposure to stressful working conditions can lead to burnout as reported by *Sonnentag et al. (1994)*. In their sample of 180 software professionals from 29 companies they found factors (e.g., lack of influence, lack of career prospects or stressors resulting from organizational policy) which can increase the risk of burnout but also approaches (e.g., improvement of team communication, challenging, interesting tasks or better career opportunities) for potential stress reduction. They measured the burnout potential with a combination of three hour long structured interviews and questionnaires.

*Fujigaki, Asakura & Haratani (1994)* looked at the stress levels of Japanese information system managers in software development. They reported 33 stressors originating from the manager role as well as from the developer role. Again, authors relied on questionnaire data (background data, work-stressor items, questions on software project management details, and measurement of stress response) to reveal those stressors (e.g., job overload, technical difficulties or work environment).

Using questionnaires, *Hyman et al. (2003)* examined the work-life balance situation of call-centre employes and software developers in the UK. Their results show that unpaid overtime is on the rise due to staff shortage or personal commitment to finish the task at hand by the end of the day. In their in-depth analysis of post-war British industry they find that, as most employes do not draw their Happiness from work, the work-life balance becomes more and more important, but harder to achieve, prompting additional stress in the lives of information and knowledge workers.

A similar approach was adopted by *Rajeswari & Anantharaman (2003)*. They again used a questionnaire with questions compiled of renowned papers from the field of research to survey Indian software professionals to identify potential stress factors. The 10 factors (fear of obsolescence, individual team interactions, client interactions, work-family interference, role overload, work culture, technical constraints, family support towards career, workload, technical risk propensity) they present in their work cover all aspects from social problems to skill related problems. This shows that these none-development related stressors will come on top of the development related problems we have identified in the introduction.

*Amin et al. (2011)* published a brief literature review of stress and what its role might be in the context of global software development. The authors talk about the importance of studying occupational stress among software engineers given their nature as intellectual workers, in particular on their activities of knowledge sharing, which they found to be most likely to be obstructed by stress-related effects. The authors conclude their review with a proposition to be further expanded by future work, that is, "In a (global software development) environment, with time zone differences, linguistic differences,

technological issues, cultural issues and lack of trust, SE occupational stress will be higher and will impede knowledge sharing." (p. 3). To our knowledge, no follow-up study exists.

*Müller & Fritz (2016)* used bio-markers to determine the difficulty of understanding a piece of code and, based on that, predict the consequences for the quality of the code. They utilized the stress indicator of heart rate changes to assess the difficulty to understand the currently examined code by a participant. They observed a statistically significant connection between bio markers of stress and the resulting code quality. In a previous study *Fritz et al. (2014)* were monitoring the brain waves of the participants. The results show that it is possible to predict the perceived difficulty of a task based on psycho-physiological markers. As the difficulty of a task can be a stressor, brain waves are an interesting indicator to measure. However, this kind of study needs specialized equipment, making it hard to be used on groups and the analysis of the results is very complex.

The influence of stress on the ability to think, memorize and concentrate has been examined by *Behroozi et al. (2018)* as stressing event. *Behroozi et al. (2018)* used technical interviews, using a whiteboard as a tool to communicate complex ideas. They used eye tracking to measure the fixations on areas of the whiteboard. The number of fixations on different areas increased under pressure indicating a lowered ability to concentrate, as the participants had to go back to previous sections more often.

*Suni Lopez, Condori-Fernández & Catala Bolos (2018)* conducted a study towards the implementation of a real-time stress-detector system based on wearable devices but following an arousal-based statistical approach as opposed to previous studies, applying machine learning for understanding emotional states and stress. The validation study adopted an ad-hoc 7-point ordinal scale for stress detection (from "not stressed" to "extremely (stressed)") and could obtain an accuracy of 80% using the arousal-based model. They found that the collaborative practices in agile might be a great source of stress. Therefore, they conducted a nationwide Swiss questionnaire on agile adoption in IT, where they asked (among other things) how agile software development influenced participants' stress at work. Stress was assessed using an ad-hoc single item, ranging from 1 (significantly less stressed) to 5 (significantly more stressed). The research analyzes correlations between the stress item and agile practices, finding that, for example, high stress levels have a negative effect on team moral.

From our examples of related work and the growing rate at which stress research in this area is conducted, we conclude that stress is a topic of interest in the computer science community.

While all previous studies contributed to our understanding, there are indications that software engineering has been avoiding using robust and validated methodology for stress detection and psychological issues in general, thus threatening the validity and reliability of studies (*Graziotin, Wang & Abrahamsson, 2015b*). Most of these studies use non-standard, ad-hoc, and non psychometrically validated questionnaires to assess the stress reaction, either by self-report or a number of questions aimed to derive the personal stress level. The use of such calls for a rather large group of participants to increase the probability to be able to report significant results. Also, the analysis of the questionnaires

leaves space for interpretation as they tend to differ from study to study and thus make it difficult to compare different studies by different researchers.

With our article and the following proposed method,we aim to critically extend the comparability of study results and hopefully overcome some of the previous limitations.

## BACKGROUND

In the following, we will provide a definition of stress and different ways to measure it. We think it is important to have this background knowledge prior to designing sound studies. Also, the information provided can help researchers who are not trained in medicine and psychology to better understand their stress target in the design phase to assess if our proposed method is adequate for their topic of research.

### Definition of stress

Stress has been viewed from medical, psychological and organizational angles resulting in many definitions. The most general definition of stress is the general adaptation syndrome (GAS) defined by *Selye (1946)*. The GAS definition can fit every scenario from personal short-time stress event to global long-term scenarios, and it was based on a formulation by *Weinert (2004)* with an organizational and workforce psychology view.

In the following we focus more closely to Weinert's explanation of GAS, as it does not dive as deep as Selye into medical details and, hence, is easier to understand for an audience without medical background.

Stress is "… an adaptive reaction to exceeding mental or physical demands of the surroundings. Adaptive, because the resilience towards those demands is individually different." (*Weinert, 2004*). *Weinert (2004)* derives that definition from these factors of stress:

1. Stress needs a physical or psychosocial trigger event.
2. Individuals react differently to that trigger event.
3. Constraints and demands are linked to the build-up of stress. Constraints and demands are, for example, deadlines or quality requirements connected to the trigger.

However, this does not mean that every event that fits the above definition necessarily affects a person.

In this context, commonly mentioned conditions for people to be affected by stress are (*Weinert, 2004*):

- The outcome or consequences of the trigger event must not be known beforehand. If the result of the stressful event is perceived as "unchangeable" it will generate no or much less stress compared to the same event with an uncertain outcome.
- The outcome or consequences of the stressor have to have an influence, either good or bad. This becomes most obvious in high-stake scenarios, for example at war, were for example, Erwin Rommel said: "Don't fight a battle if you don't gain anything by winning." (*Rommel & Pimlott, 2014*).

For better understanding, let us make an example related to software development:

A software product is due to be released to an important customer. The future of the company depends on this commercial operation because funds are running out. It is not known yet whether or not the customer will buy the product in the current state.

The **trigger event** in our example is the prompt release of the product (deadline). The **consequences** of the stressor are not known, as it is not clear if the product will be sold, at what price and if this sale will keep the company financially afloat. The **outcome** is important to the developer because his/her job might depend on it.

Each person in the company might experience a **different level of stress** connected to this scenario based, for example, on their **individual judgement** of the ease of finding a new job if the project is not successful and the company goes bankrupt. If a person has already taken mental dismissal and is sure to find a new job or already has a new job he/she might not experience any stress at all.

The GAS can be used to model every stress interaction in general. As in the example above, it can be used to view the impact of stress (the critical release of the software product) on the company as a whole. A more narrow focus is the theory of *Lazarus (1966)* which refines the idea of Selye.

By narrowing the reaction to a stressor to the cognitive processes active when dealing with stress (transactional or cognitive stress theory), this model is closer to the situation of knowledge workers such as software developers than the generalized model of Selye. In the transactional model, if a situation is assessed to be a strain, the situation can be considered as already harming, potentially harming (threatening) or as a risky, but worthwhile, challenge. The assessment and the progress of the situation based on the personal resources and the possible solutions for coping with the stressor can change over time. Based on these possibilities an actual reaction is provoked. This reaction to the stressor is, in the best case, a direct action to solve the stressful situation (fight) or, in less favorable constellations, evasion (flight) of the situation or other coping strategies (e.g., trivialization).

This cycle of assessment based on the changing surroundings and continuous evaluation of coping strategies will be repeated until an appropriate coping strategy is found, which ends the stressing encounter. If no fitting coping strategy can be found, the person is either blocked by the continuing search cycle or through the application of unfitting solutions, with a high usage of resources, the problem is gradually eroded until it can be completely overcome.

Let us again imagine a software engineering example for this stress model:

A new member of a development team has been assigned to the first task in the project. It is a non-trivial part of the system to be implemented and it is the developer's chance to prove his/her worth to the team.

After the situation was found to be stressful, a second assessment of the problem reveals that the new member feels a lack of skills needed to finish the given task properly

(situation assessed to be threatening). Despite that feeling he/she still starts working on the problem (coping). The new team member is working on the problem and his/her knowledge and skills increase as he/she is going and the assessment of the situation may change to "risky but worthwhile". The assessment can change again in the course of action, for example if the new member encounters a problem which can not be fixed easily. The assessment can then again rise to threatening or even harmful, depending on the personal resources available. Still the assessment will continue until the situation is solved one way or another.

It is important to keep these definitions of stress in mind when designing a study or experiment which aims to measure stress because we need to be aware of the stress generation, which is still not fully understood (*Noack et al., 2019*). Most of the time we will already have a stressor we want to take a look at (e.g., project deadlines), but to keep that stressor as free from other influences as possible, and for a correct interpretation of the results later on, we need to look at potential constraints and demands which might not affect all participants in the same way. We also have to find a way to make the participant care about the outcome of the stressor if it is an artificially created stressor. There are some commonly used ways to induce stress in experiments like the Trier Social Stress Test (*Kirschbaum, 2015*) which uses social evaluation and cognitive demands to generate stress. Another way can be to create a competitive scenario in which participants can earn a reward based on their results on a task compared to the other participants.

## Effects of stress

To understand the ways to measure stress, we need to know the basic reactions of the body and mind to stress. A frequently cited summary of the general effects of stress has been written by *Kaufmann, Pornschlegel & Udris (1982)*. Somatic psychological short-time effects include increased heart rate, raised blood pressure and adrenalin discharge. The personal psychological effects might include a feeling of strain, frustration, anger, fatigue, monotony and saturation. The individual behavior can suffer from performance fluctuation, decline of concentration ability and bad sensorimotor coordination, leading to an increased error rate. Medium and long term stress exposure can lead to psychological problems such as dissatisfaction, resignation and depression and general psychosomatic ailment and disease. The negative effect of medium to long-term exposure to stress on the behavior can include increased consumption of nicotine, alcohol or drugs as well as absenteeism (sick days) on individual level and conflicts, quarrels, general aggression against others and withdrawal (isolation) at and outside of work on a social level. Even short-term stress can lead to unpleasant side effects, which are especially harmful for communication and cooperation-dependent work like software development. Since 1982 when *Kaufmann, Pornschlegel & Udris (1982)* summarized the effects, more research has been conducted supporting and extending the understanding of stress effects mentioned by them. We will go into more detail below.
## Physical reactions

[1] Where not explicitly cited, the reference is taken from the textbook "Biologische Psychologie" by *Birbaumer & Schmidt (2010).*

[1] The physical reactions of the human body can be explained by the survival needs of our ancestors. It is often called the "fight or flight" reflex, first introduced by *Cannon (1929)*. In stressful situations in prehistoric times, for example, the encounter with a predator, the body needs energy to fight or to run away (flight), so it releases adrenaline to the blood stream, ordering the endocrine system to start providing chemical energy. *Noto et al. (2005)* described the effects of stress on the endocrine system focusing on cortisol and alphaamylase. In short, under stress cortisol and alphaamylase concentration increases providing additional energy to the organism (e.g., increasing the blood sugar). Both these effects help the body to release chemical energy (e.g., sugar) to the blood stream. These effects are traceable in saliva. The heart rate increases (*Vrijkotte, Van Doornen & De Geus, 2000*) in stressful situations to transport the mentioned substances faster through the body as well as to provide more oxygen which is needed to utilize the chemical energy carriers freed by the cortisol and the alphaamylase. Due to the increased body activity, sweat can break out, changing the dialectical conductivity of the skin as a result. Sweat might also be a result of physical work which is considered a form of stress for the body. Stressful exhaustion might lead to involuntary muscle contraction (tremors). But not only physical stress can lead to involuntary muscle contraction. Getting tired as a result of work/stress leads to increased blinking.

These are short-term reactions. If these-short term effects are prolonged, they can have negative effects on the human body. Commonly mentioned effects of long-term stress on the human body are: high blood pressure, high cholesterol values and heart diseases (*Weinert, 2004*; *Kaufmann, Pornschlegel & Udris, 1982*; *Richter & Hacker, 1998*). The physical reactions of the human body to stressful events can change based on the demands and resources (e.g., a more muscular person might endure physically demanding work longer than the average person).

Most of the reactions to stress are internal, steered by the hormone system. The increase of these chemical messengers can be utilized as a measurement as they remain in the blood and also in saliva and urine for some time.

## Psychological reactions

If the stressing situation prevails, it has negative short and long-term effects not only on the body but also on mental health. Mental health problems related to stress are on the rise in the modern world as shown by *Lademann, Mertesacker & Gebhardt (2006)* in their analysis of the sick notes submitted to health insurances or in the stress report for Germany (*Lohmann-Haislah, 2013*) which gives a yearly overview of the stress situation of the German workforce.

*Cohen (1980)* wrote a summary of the research on stress effects so far. Among other topics, he wrote about after-effects on the social behavior of stress plagued persons. He reported about various experiments where the participants previously exposed to stress showed significantly less helpfulness and empathy as well as higher levels of aggression towards other people. Amongst others, *Weinert (2004)* listed as psychological effects of stress (similar to the definitions seen in "Effects of Stress"): poor concentration, difficulty

making decisions, obliviousness, thought blockades and burnout as well as subjectively experienced anxiety and lethargy. The authors focused on the negative aspects of prolonged exposure, but should not forget that short-term stress also has positive effects, like increased motivation and energy as discussed by *Folkman (2008)*.

Also, both stress reactions are subjected to individual perception of stress. The perception of stress can be modified by, for example, changes at the workplace such as placement of tools or changing working positions. It is also bound to personal stress resilience, which can be genetic or mental, but can be increased by focused training.

Stress is a multi-faceted phenomenon which makes it challenging to measure. There are many methods proposed by researchers from distinct science branches and interdisciplinary research, for example by *Kanner et al. (1981)*, *Lazarus (1990)*, *Cohen, Janicki-Deverts & Miller (2007)* and *Plarre et al. (2011)*.

## STRESS MEASUREMENT TECHNIQUES

As we have seen in the previous section, stress is a multifaceted construct that can be defined in different ways according to different disciplines. Following the work by *Cohen, Kessler & Gordon (1997)*, the measurement of stress is split into mainly using psychological measurements, such as questionnaires, and mainly using measurements of biological markers, such as hormone levels and psycho-physiological reactions (*Vrijkotte, Van Doornen & De Geus, 2000*).

### Psychological measurements

Many different variations of psychological measurements of stress have been discussed over time (*Cohen, Kamarck & Mermelstein, 1983*; *Peacock & Wong, 1990*; *Kanner et al., 1981*). Psychological measurements can be grouped into two classes: those which assess long-term stress (e.g., Life Changing Events by *Rahe (1977)*) and short-term stress (e.g., Emotional Self Rating by *Schneider et al. (1994)*) based on self report and concrete situations, and those which look at more global coping strategies of participant who are not directly connected to a specific trigger (e.g., Stressverarbeitungsfragebogen by *Janke, Erdmann & Boucsein (1984)*).

The long-term assessment strategies are used to investigate the impact of stress on the well-being or health of individuals. This method uses interviews or tests that should represent the overall picture of the stressful events in someone's life, for example, the death of a loved one or job loss.

The short-term assessment strategies try to assess the stress experienced for some minutes up to an entire day by at least two questionnaires or interviews, ideally before and after the stressful event. The items used in these tests range from a plain assessment by the participant on the momentary stress level on a scale from 1 to 5 to more episodic tales of events including the points of why this event was found to be stressing, how severe and long the stress was felt. This kind of data collection is used regularly in psychology as well as medical research and is commonly considered as reliable given good psychometric properties.

An example from the software engineering point of view to distinguish between long and short-term term stress assessment could be the difference between keeping track of a whole development project, taking a sample each day and assessing the stress of a single 4-h programing session.

The other class, aiming at looking at stress coping strategies, uses questionnaires that ask general questions like "How do you handle stressful situations?" or "What kind of situations do stress you?". This can also be done over a period of time, to gather a stress profile, by repeating these question over the day.

Stress also affects concentration and memory (*Weinert, 2004*) because of the cognitive resources needed to cope with the increased stress. This use of cognitive resources is called cognitive load. Cognitive load can be measured by testing concentration and memory of participants using tests like the *N*-Back-test (*Gevins & Cutillo, 1993*).

Stress can also have a negative effect on a person's mood, leading to frustration and anger at short-term and, at worst case, to a depression at long-term exposure, as mentioned in "Effects of Stress". Mood can be assessed by questionnaires like PANAS (Positive and Negative Affect Scale, *Watson, Clark & Tellegen (1988)*) or ESR (Emotional Self Rating scale, *Schneider et al. (1994)*). All these questionnaire based approaches can suffer from common issues and biases when working with humans participants (*King & Bruner, 2000*; *Paulhus, 1991*; *Schwarz, 1999*):

- Honesty/Image management: The problem how the participant wants to present him/herself and what he/she is willing to reveal about him/herself.
- Social desirability: The distortion of answers by the participants because they feel like some answers are more socially acceptable than others.
- Introspective ability: Concerns to which degree the participant is able to understand him/herself.
- Understanding: As human language is not perfect and needs to be interpreted, formulations can be understood differently.
- Rating scales: The nuances of ratings can be interpreted differently.
- Response bias: The individual's tendency to respond in a certain way, regardless of the actual evidence they are assessing.

Psychological tests have ways to balance such issues by providing control answers which show contradictions. Despite that, it is always desirable to back up these results with biological factors which are much harder to be influenced by involuntary effects and biases.

## Biological markers

The biological factors available to us and mentioned in literature to measure stress are heart frequency, blood pressure, electrical conductivity of the skin, activity of muscles and hormone levels (*Birbaumer & Schmidt, 2010*). The assessment of these factors is not easy, as the difference between a normal phase of strain and abnormal stress needs to be considered and not every factor is applicable to each form of perceived stress (physiological, emotional, mental).

### Skin conductance

The skin conductance is measured whenever the differential or changing impact of stress needs to be assessed (*Andreassi, 2013*). Electrodes have to be attached to the participants non-dominant hand. This approach has two main disadvantages: first, it might introduce additional stress as the electrodes are a constant reminder for the participant that they are being monitored and, in a typical case of software engineering research, it is not applicable to larger groups in parallel because of the need of multiple devices for measurement and medical trained personal for the correct application of the electrodes. Sensors for measuring the sweating can be used if only a difference between calm and stressed states is of interest and no finer assessment is required.

### Heart rate

The measurement of heart rate and blood pressure can be a sensitive tool to measure mental stress (*Hjortskov et al., 2004*) but it suffers from the same disadvantages as the measurement of skin conductance: the invasion of personal space and the in-feasibility to extend it to large groups, for the same reasons as mentioned for conductivity of skin.

### Muscle activity

We base our discussion of muscle activity on the works by *Boucsein (1991*, *1993)*. Muscle activity can be measured, among other ways, via eye blink frequency, eye movement or mydriasis (size of the pupil) using an electromyogramm or observing muscle tremors. The difficulty to obtain the measurements and the expressiveness of these measures varies widely. While blinking frequency, eye movement and size of the pupil are relatively easy to measure for individuals, the results are heavily dependent on the task given and are only significant in combination with other measurements. Tremors can be observed optically, but are mostly a sign of physical stress which is not a form of stress normally induced by computer work and so it is out of the scope of this work.
The electromyogramm is a good method to measure muscle activity using electrodes similar to the one used to measure skin conductance. This again is more meaningful for the research on physical stress which the average software developer is not likely to experience in an extended amount in day to day work. Besides the low relevance for software engineering, the methods to measure muscle activity suffer from the same disadvantages as mentioned for skin conductance.

### Hormone level

The usefulness of hormones for stress assessment has been shown by *Goldstein (1995)* and *Noto et al. (2005)*. The hormone levels of *cortisol* and the protein *alpha amylase* (*Nater et al., 2005*) which indicates the utilization of blood sugar are found in saliva samples (*Chiappelli, Iribarren & Prolo, 2006*; *Kirschbaum & Hellhammer, 1994*; *Chatterton et al., 1996*; *Reinhardt et al., 2012*). Saliva can be gathered and stored for a couple of days in plain plastic tubes, without any specialized equipment or sophisticated cooling mechanism. If the samples are to be stored for a longer time, they have to be frozen to avoid degradation. Samples can be sent to a laboratory for analysis via postal service. The laboratory will use an analysis kit and return the analysis results. The results can be

interpreted using basic statistical tests. Support by medical trained scientists improves the quality of the conclusions drawn from the results as they might be able to explain possible outliers originating from health problems of the participants or medical drugs used by participants.

Hence, we opt for the latter biological family of stress assessment which we pair with validated psychological assessment. We propose a methodology in the following.

## PROPOSED METHOD FOR STRESS MEASUREMENT

The method we propose, inspired by *Kirschbaum & Hellhammer (1994)*, *Van Eck et al. (1996)*, and *Strahler et al. (2017)*, is applicable to controlled experiments in software engineering which aim to examine the effects of stress. It consists of measurements of psychological markers, measurements of biological markers, and a description of sequences and details of the measurements. The measurement tools for the psychological and biological markers have been selected in cooperation with psychologists, have been used in a massive amount of studies (e.g., *Hellhammer, Wüst & Kudielka (2009)*, *Khalfa et al. (2003)*, *Thompson et al. (2012)* on hormone usage, e.g., *Pressman & Cohen (2005)*, e.g., *Jennett et al. (2008)* on PANAS and *Weiss, Salloum & Schneider (1999)*, *Schneider et al. (1999)* on ESR), and are commonly agreed to be reliable and valid.

### Measurement tools

We propose to assess stress, emotional state and cognitive load of participants, where the first factor is observed both from a psychological and a biological perspective. Figure 1 describes in greater detail the constructs under study and the related measurement instruments.

We assessed stress using biological markers through saliva samples. In particular, we measured cortisol and alphaamylase, as the saliva samples are easy to gather and the hormone/enzyme levels in it are easy to measure by any laboratory specialized in hormone analyses. We propose PANAS (Positive and Negative Affect Scale, *Watson, Clark & Tellegen (1988)*) and ESR (Emotional Self Rating scale, *Schneider et al. (1994)*) for the psychological stress and emotional stress markers as self-ratings have a good time-to-information ratio. PANAS and ESR are used to assess the participants emotional state and current mood, as increase in stress has a negative effect on mood (see "Effects of Stress"). We extended PANAS with questions asking about pre-existing stress levels and possible previous knowledge and skills related to the task to be done in the experiment, because previous knowledge can have an impact on coping strategies of the individual and have an impact on stress generation. With these extensions our PANAS scores for the positive factors ranges from 0 to 50 and the negative factors from 0 to 55.

Cognitive load (see also "Psychological Measurements" on Psychological Measurements) is conveniently operationalized and measured by the *N*-Back test as implemented by the computerized PEBL Psychological Test Battery (*Mueller & Piper, 2014*). PEBL allows the data to be gathered automatically and exported as datasets. *N*-Back challenges participants to memorize letters and their position in a sequence of letters which are shown one letter at a time. With a *N*-Back test, participants have to press a key if a

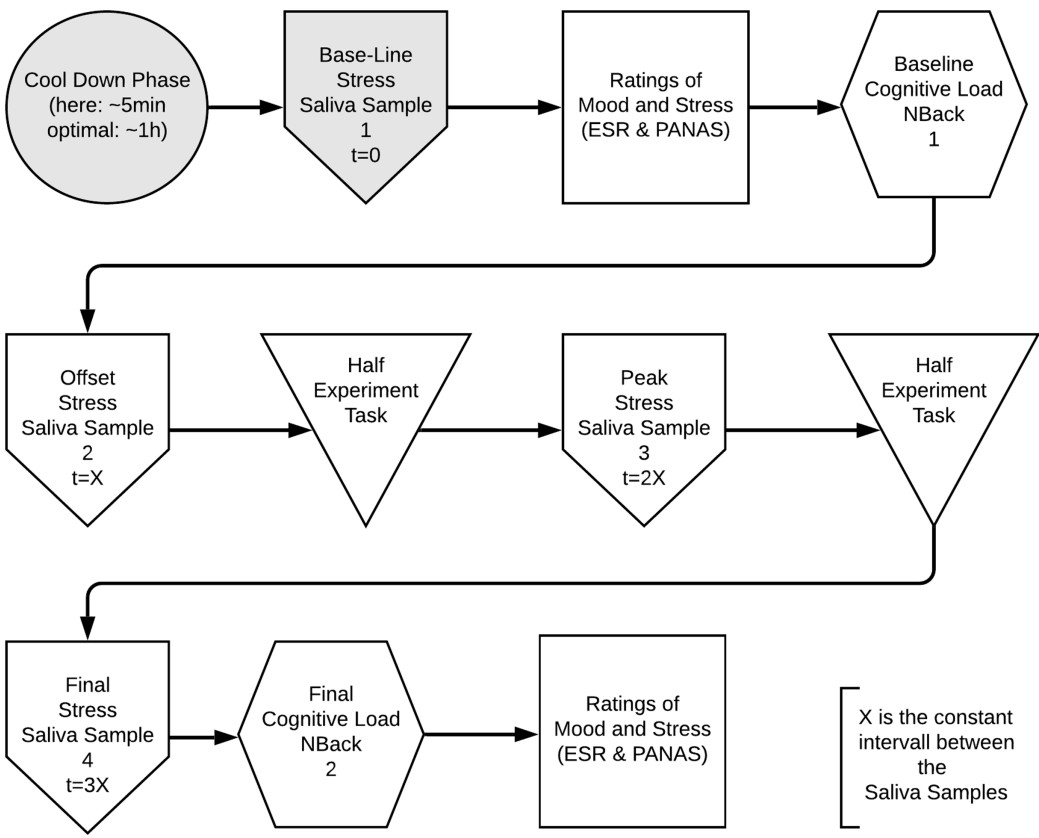

If it is of interest to assess stress recovery, we recommend to add another stress and saliva sample 40 to 120 minutes after the stressor onset.

**Figure 1 Our proposal for a robust and sound experimental design to assess causes and consequences of stress in software engineering.** The grayed-out activities were omitted in the final design iteration.

letter has been repeated $n$ steps back. For example, for the letter sequence {X, X, X, V, X, Y, Y} and a 1-Back test, the key should be pressed at (1 for key press, 0 for no key press) {0, 1, 1, 0, 0, 0, 1}. The correct key presses for a 2-Back test would be {0, 0, 1, 0, 1, 0, 0 }. The hit/miss ratio and reaction time indicates how much cognitive capacity is left to work with. In order to reduce the learning effect when the test is used multiple times, there are variations of this test (e.g., not letters, but positions in a $3 \times 3$ square are to be correctly remembered).

Besides demographic data, a study should also collect control-variables that might influence the stress reaction, such as pre-existing mental conditions and medications (e.g., birth control pill). Also, as this might have an increasing effect on personal stress, we asked how satisfied the participants were with their decision of life and work situation alongside the demographic questions. The data was gathered anonymously and linked together only through an anonymous identifier.

## Measurement sequence

The first step in implementing our proposed method is to decide on the frequency and placement of the cortisol and alphaamylase measurements. The decision should take into consideration that, while more measurement points in a shorter period of time will provide a more detailed picture of the stress reaction to the topic under observation, it also will affect the stress generation itself. The shortest cycle is also limited by a few parameters. Participants only have a limited ability to generate enough saliva. Too frequent disruptions might be perceived as a nuisance, increase the generated stress, and might have an additional negative effect on the topic under research or even cover the stress reaction under observation. Also, the hormone system reacts in the range of minutes to hours whereas the nervous transmission reacts in milliseconds (cortisol peeks around 15–20 min after stress onset (*Kirschbaum & Hellhammer, 1994*)). If the stress-inducing task is too short (less than 10 min) a second sample should be gathered to increase the chance to include the peak or other measurements (e.g., heart rate or skin conductivity) can be used, but with the impediments stated above. As cortisol has a daily cycle, the gathering of the saliva samples also needs to be planned in a way that all participants are assessed at the same time to generate comparable results. In order to add noise to the cortisol and alphaamylase measurements, it is important to instruct the participants to not consume beverages containing sugar and not to eat or smoke one hour before the saliva samples are gathered.

The times when the personal stress and emotional state are assessed have to be adjusted as well. As it takes a non-trivial amount of time to fill in the questionnaires, the personal and emotional assessment should happen at the beginning of the study to determine a measurement baseline, then at appropriate times in the course of the study and at the end of it. In the case of short periods of saliva sample measurements it is sufficient to have the personal stress levels and emotional state measured at the start and end of a task. If the topic under research contains longer breaks we advise to use the questionnaire measurement at the start and end of the breaks. For long term studies, we advise to use the self-assessment via questionnaires at least twice a day, possibly at the beginning and end of the work day. By doing so, the influence of stress generated outside the object under research can be identified and taken into account when looking at the stress reaction.

In our case, the measurement of the cognitive load happens after a substantial part of the task under research and before the questionnaires assessing the personal stress level and mood, so the cognitive load is only influenced by the task.

As with the questionnaires, for long-term studies, the *N*-Back should be used at the start and end of the working period.

We illustrate the entire design process in Fig. 1. It represents the process as it was derived from our literature review and consultations with colleagues from the psychological and medical fields. Participants face a short cool-down period (approx. 5 min) of no activities, for controlling purposes, during which most of their markers stabilize. We then instruct participants about the experiments goals. After that, the participants sign a consent form (not present in Fig. 1). We collect a first sample of

saliva, which sets the baseline stress value of participants. After the baseline stress measurement, participants fill in demographic questionnaires. Following the demographic questionnaires, we start assessing the baseline mood and perceived stress with ESR and PANAS. Participants then face the first computerized task, that is the *N*-Back test, for assessing the baseline cognitive load. As the *N*-Back test might be stressful to participants, we take a second saliva sample to be able to monitor the stress build-up for the task under observation later. The "stress" task for the experimental and control groups can then begin.

The length and amount of the experimental task parts (Fig. 1 shows two parts) should be dictated by the decision on the saliva sample interval. In our case, we take a third saliva sample at around half of the time planned for task solving. The stress markers cortisol and alphaamylase should be peaking here as the endocrine system has had enough time to respond to the initial stress trigger of the task (cortisol peeks around 15–20 min after stress onset (*Kirschbaum & Hellhammer, 1994*)).

We take a fourth and final saliva sample at the end of the experimental task. Finally, participants do the computerized cognitive load test once more, to measure the available cognitive resources left after the experimental task. A final set of ratings of mood and perceived stress follows. The last two activities are inverted compared to those before the task, as the cognitive load score should not be influenced by the questions for rating mood and self-assessment.

Longer studies performed over several days have a similar design to Fig. 1, varying only in the intervals of measurements as mentioned above. If the assessment of the recovery from the stressor is of interest, a measurement of all relevant indicators (cognitive load, mood, perceived stress, hormone/enzyme levels,…) should be done 40–120 min after the stressor onset. In the remainder of this paper, we report on two studies that allowed us to implement and refine the overall research design.

## TWO STUDIES IMPLEMENTING THE PROPOSED METHOD

In the following, we present two studies, the first being a pilot, implementing our proposed method for stress assessment. The lessons learned from the application of this method helped us with its refinement.

The purpose of our studies was to analyze stress reduction effects, cognitive load reduction, mood improvement, and software quality enhancement of visual and user experience changes to the static analysis tool FindBugs. The control group used the latest version of FindBugs. The experiment group used a version of FindBugs which was modified following the Salutogenesis principles. Salutogenesis is a well-being theory, based on comprehensibility, meaningfulness and manageability to explain perceived stress or to help cope with it by changing these variables. We have previously proposed this for enhancing the interaction of software developers with their tooling (see *Ostberg & Wagner (2016)* and *Ostberg et al. (2017)* for details on Salutogenesis). By design, all tasks required a considerable effort and it was not possible to finish them in the time given. The varying difficulty of the subtasks does not require the participants to meet a certain

level of skill, but some basic understanding of programming, yet still provides enough of a challenge for the advanced participant.

## Pilot study 1

The pilot study implemented the method of Fig. 1 in its entirety. We only added a set of questions aiming at self-efficacy (*Bandura, 1997*; *Jerusalem & Schwarzer, 1999*; *Kogler et al., 2017*) to the psychological test phases, which allowed us to assess the individual stress resilience of the participants.

For the pilot we recruited 43 volunteers from the MSc course "Software Quality Assurance and Maintenance" at the University of Stuttgart. Students were in their 2nd or 3rd semester of studies. None of them reported any medical conditions interfering with the stress measurement. We were aiming for a high number of participants even in the pilot as we wanted to evaluate how easily large groups can be assessed with our chosen methods.

To induce stress, the participants were told that a list with the results of their work on the code (number of correct fixes) would be made public, so every participant could see how well he or she did, compared to the other participants. We never delivered on our threat, for obvious ethical reasons, but the stress was induced by our statements.

We split the work phase into two parts of 25 min and 20 min, the first part was a bit longer in order to compensate for the time needed to get into the task.

To avoid learning effects, the second *N*-Back test used positions in squares rather than letters.

### Results

Unfortunately, we faced some data loss due to failing hard drives. Boxplots in Figs. 2–4 shows the data we were able to retain. Figure 2 shows the distribution of the PANAS values. From bottom to top we have the sum of the positive factors before the task, the positive factors after the task, the negative factors before the task and the negative factors after the task. The top four plots show the difference between pre and post PANAS scores. This shows the progression of the emotional state of the participants. Figure 3 shows the number of correct responses to the *N*-Back stimulus in percent and Fig. 4 shows the reaction time in milliseconds to the *N*-Back stimulus, both pre and post the task. The values for correctness and reaction time are connected, so the participants for both values are the same.

Of the PANAS and ESR questionnaires, 10 out of 20 from the experiment group and 11 out of 23 from the control group were usable. The emotional state based on the PANAS values is calculated by summing up the positive effects and subtracting the sum of the negative effects. To see the emotional change, we calculated these values pre and post-work phase. The difference of these values indicates the emotional change. Performing a repeated measures (rm) ANOVA with time (pre, post) and valence (pos, neg) as within-subject factors and group as between-subjects factor revealed no significant effect of time ($F(1,18) = 1.540$, $p = 0.230$), a significant valence effect ($F(1,18) = 4.615$, $p = 0.046$, part-eta-sq. = 0.204) with higher values in the positive valence than negative valence,

**PANAS Results**

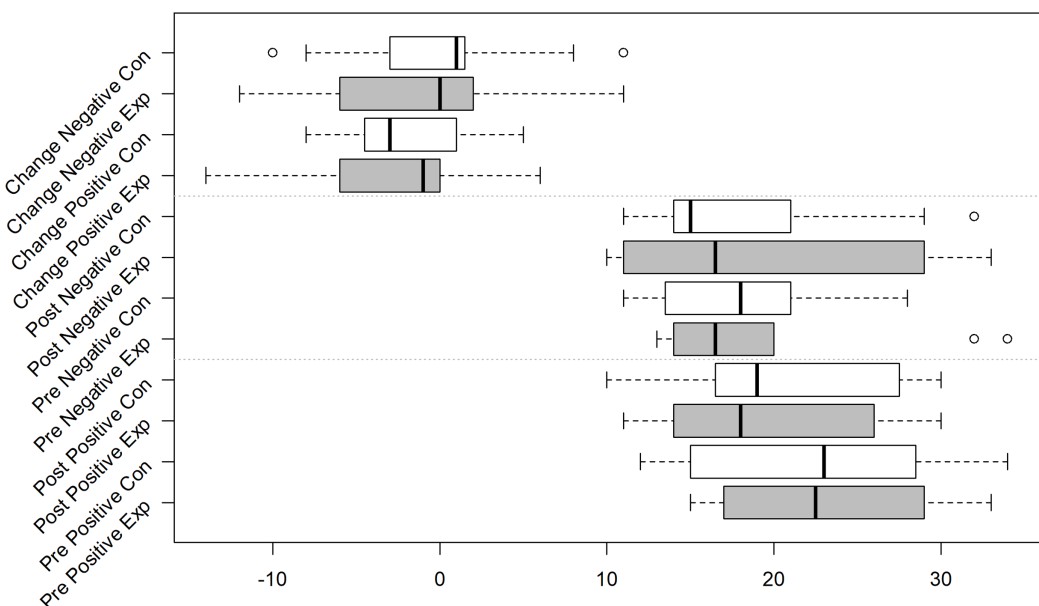

**Figure 2 PANAS Pos(itive) and Neg(ative) Mood Indicators Exp(erimental) Group (Sample size = 10) and Con(trol) Group (Sample size = 11).**

**NBack Correctness Results**

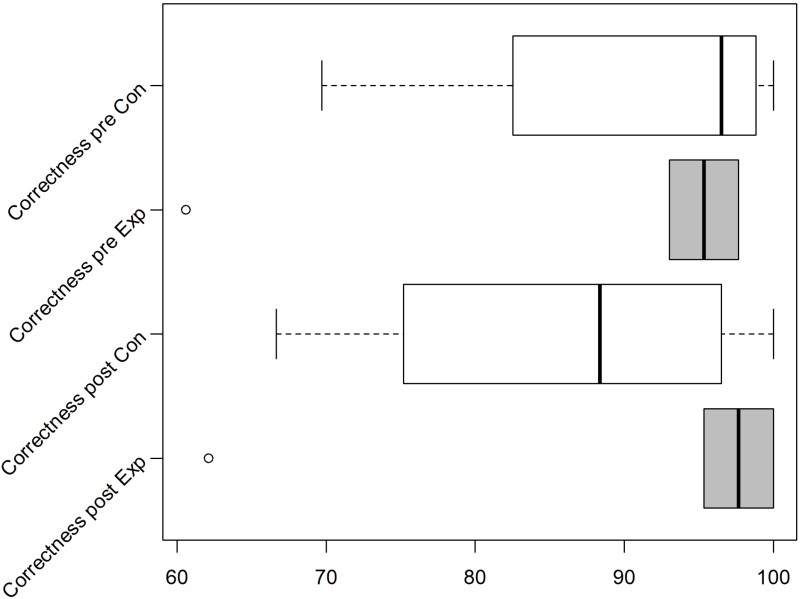

**Figure 3 N-Back correctness for the Exp(eriment) Group (Sample size = 6) and Con(trol) Group (Sample size = 4).**

**NBack Response Time Results**

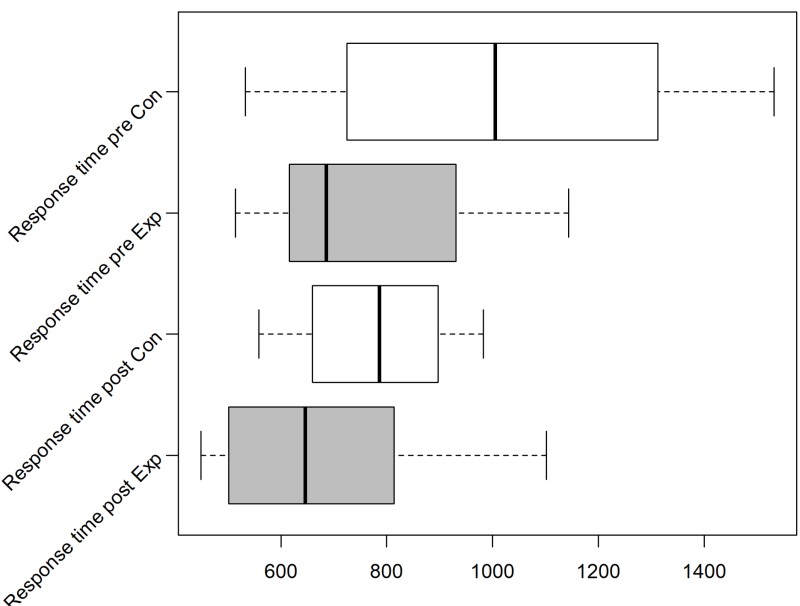

**Figure 4** *N*-Back reaction time for the Exp(eriment) Group (Sample size = 6) and Con(trol) Group (Sample size = 4).               

**Table 1 Pre and post-experiment task values for ESR factors, experiment group, pilot, sample size = 10.**

|  | Anger | Disgust | Happiness | Sadness | Surprise | Fear | Stress |
|---|---|---|---|---|---|---|---|
| ESR pre experiment group |  |  |  |  |  |  |  |
| Mean | 1.90 | 1.40 | 1.80 | 1.50 | 1.90 | 1.30 | 2.50 |
| Standard deviation | 0.88 | 0.84 | 0.92 | 1.08 | 0.99 | 0.67 | 0.85 |
| Median | 2.00 | 1.00 | 1.50 | 1.00 | 1.50 | 1.00 | 2.00 |
| ESR post experiment group |  |  |  |  |  |  |  |
| Mean | 2.3 | 1.3 | 1.6 | 1.6 | 1.9 | 1.4 | 2.7 |
| Standard deviation | 1.06 | 0.67 | 0.84 | 1.07 | 1.37 | 0.84 | 0.95 |
| Median | 2.5 | 1 | 1 | 1 | 1 | 1 | 3 |

and no significant group effect ($F(1,18) = 0.192$, $p = 0.667$). Moreover, no interaction reached significance (all $p > 0.151$).

The data of the ESR (see also Tables 1 and 2) does also not show statistically significant group differences for the emotion (all $p > 0.395$) and stress ($p = 0.342$) ratings.

Table 3 reports all values we gathered for cortisol and alphaamylase. See Fig. 1 for the location of the various measurements in the study progress. The sample size refers to the actual sample size used for calculations at this step, as the lab analysis reported invalid values for some steps/participants, probably due to not enough saliva samples left as they had to rerun some of the analysis. From the numbers we can see a build up of a

**Table 2 Pre and post-experiment task values for ESR factors, control group, pilot, sample size = 11.**

|  | Anger | Disgust | Happiness | Sadness | Surprise | Fear | Stress |
|---|---|---|---|---|---|---|---|
| ESR pre control group |  |  |  |  |  |  |  |
| Mean | 2.09 | 2.18 | 1.82 | 1 | 1.91 | 1.36 | 2.18 |
| Standard Deviation | 1.22 | 1.33 | 0.75 | 0 | 0.83 | 0.92 | 1.17 |
| Median | 2 | 2 | 2 | 1 | 2 | 1 | 2 |
| ESR post control group |  |  |  |  |  |  |  |
| Mean | 2.09 | 2 | 2 | 1.18 | 1.18 | 1.23 | 2.73 |
| Standard Deviation | 1.38 | 1.34 | 1 | 0.4 | 0.98 | 0.65 | 1.27 |
| Median | 2 | 1 | 2 | 1 | 2 | 1 | 3 |

**Table 3 Cortisol and alpha-amylase scores, pilot.**

|  | Experiment | Control | Experiment | Control |
|---|---|---|---|---|
|  | Baseline cortisol (pg/ml) |  | Offset cortisol (pg/ml) |  |
| Mean | 6.13 | 7.18 | 5.63 | 7.3 |
| Deviation | 4.41 | 4.24 | 3.93 | 5.15 |
| Median | 4.74 | 7.2 | 4.71 | 6.17 |
| Sample size | 14 | 22 | 17 | 22 |
|  | Peak cortisol (pg/ml) |  | Final cortisol (pg/ml) |  |
| Mean | 6.69 | 7.11 | 5.38 | 4.5 |
| Deviation | 4.05 | 5.09 | 3.73 | 2.76 |
| Median | 5.77 | 5.81 | 5.39 | 4.2 |
| Sample Size | 15 | 18 | 11 | 15 |
|  | Baseline alphaamylase |  | Offset alphaamylase (U/l) |  |
| Mean | 44.33 | 55.29 | 57.15 | 87.1 |
| Deviation | 47.77 | 83.16 | 86.32 | 81.66 |
| Median | 19.63 | 18.76 | 20.83 | 50.24 |
| Sample size | 14 | 19 | 14 | 21 |
|  | Peak alphaamylase (U/l) |  | Final alphaamylase (U/l) |  |
| Mean | 62.39 | 44.73 | 84.08 | 47.35 |
| Deviation | 78.67 | 56.75 | 106.23 | 54.74 |
| Median | 18 | 18.95 | 18.76 | 20.39 |
| Sample size | 10 | 17 | 11 | 13 |

hormone stress response with a slightly later peak in alpahamylase for the experiment group.

The statistical tests (Willcoxon $U$ for $\alpha = 0.05$ for cortisol (C1 = 0.29, C2 = 0.31, C3 = 0.54, C4 = 0.61), Cliff's delta for cortisol (C1 = −0.068, C2 = −0.76, C3 = −0.3, C4 = −0.564), $t$-Test for $\alpha = 0.05$ for alphaamylase (A1 = 0.64, A2 = 0.31, A3 = 0.54, A4 = 0.36) Cliff's delta for alpha amylase (A1 = −0.42, A2 = −0.18, A3 = −0.57, A4 = −0.61)), reveal no significant differences between experiment and control-group. A rmANOVA

revealed no significant time effect ($F(3,36) = 1.096$, $p = 0.363$), no significant group effect ($F(1,12) = 0.515$, $p = 0.487$) and no significant group*time interaction ($F(3,36) = 0.278$, $p = 0.788$). Similar to the results on cortisol, rmANOVA with alphaamylase levels indicated no significant time effect ($F(3,33) = 0.490$, $p = 0.691$), no significant group effect ($F(1,11) = 0.861$, $p = 0.373$) and no significant time*group interaction ($F(3,33) = 0.443$, $p = 0.636$). The raw data of the hormone levels can be found in the appendix in Tables A1 and A2.

The increasing effect size might imply that the differences between the control and experiment groups would grow more visible if the experiment would progress longer and if we had had more usable data points. It is also possible that a greater stress induction will lead to a more visible effect.

Due to various problems at the time of the experiment's execution, we were only able to gather 4 usable data sets (correctness/reaction time for pre and post task) for the control group and 6 data sets for the experiment group for the *N*-Back test. For these data sets, the other data (PANAS, ESR and hormone/protein levels) is also available. In the pilot we see an improvement of correctness and reaction time for the experiment group (see Median in Boxplot Figs. 3 and 4), while the correctness for the control group decreases. Still, analyzing the effect of intervention on cognitive performance, the rmANOVA with the within-subject factor time (pre, post) and between-subjects factor group revealed no significant time effect ($F(1,8) = 0.479$, $p = 0.509$), no significant group effect ($F(1,18) = 0.091$, $p = 0.770$), and no significant time*group interaction ($F(1,8) = 3.391$, $p = 0.103$) emerged.

### *Conclusion*

Despite the data being inconclusive, the pilot experiment showed the feasibility of the design. The saliva samples were easy to collect for both, the participants as well as for the researchers, and indications given by the physical measurements match the indications of the psychological measurements. We used the lessons learned to make some changes to the study design which we will discuss next.

## Study 2

In the design of the second study the grayed-out parts of Fig. 1 are removed. This represents our changes based on the lessons learned of the pilot. We removed the initial cool-down factor and the first saliva sample. For the sake of consistency with the figure, we still call our new first saliva sample the offset stress, but we consider that measurement step our new baseline stress. The pilot test did not suggest potential differences in the measured levels within baseline stress and between baseline and offset stress, so we assume that the psychological test and *N*-Back session do not induce any significant stress to software developers. As a positive side effect, this elimination of one of the saliva samples reduces the time and money needed for the laboratory analysis.

We also reduced the amount of questionnaires in the psychological test phases by removing the resilience questions. The value of these question items did not help in

explaining any of the effects analyzed. During our observation and in post-experiment statements, participants seemed to be most irritated by the resilience questions.

We also changed the way we induce stress to the participants in the hope that the changed procedure would show an increased reaction. First we stimulated the participants with a hardly reachable goal for the work on the software. We told them that previous participants had done a minimum of 35 fixes for problems highlighted by FindBugs in 3 different categories and that these were the low achievers. Second, as the groups were much smaller (3–5 participants per session) we could recreate an effect similar to the TSST (*Kirschbaum, 2015*) effect, by simply having evaluators in the room monitoring the work of the participants and pretending to write down remarks on their notepad from time to time with muttering disapproval.

For this experiment we were able to recruit 32 participants from the bachelor study course "Introduction to Software Engineering". The participants were randomly distributed over 7 dates, 3 for the control group and 4 for the experiment group, resulting in 17 participants in the experiment group and 15 participants in the control group. Again, as in the pilot, the experiment group used our enhanced tool, while the control group used the original tool. None of the participants reported any medical conditions interfering with the stress measurement.

### Results

The results of the PANAS questions (see Boxplot in Fig. 5, missing samples either did not finish or did not fill in questionnaires fully, see "Pilot Study 1" for description of layout) show that both groups experienced a decrease in mood, but the decrease was steeper for the control group (Median for positive factors declines by 6, median for negative factors increases by 3) than for the experiment group (Median for positive factors declines by 2, median for negative factors increases by 2). Still, the range of mood change for both groups is spread over a wide spectrum, so the results are not statistically significant (Wilcox $p = 0.6$ for $\alpha = 0.05$) and the effect size (Cliff's Delta: $-0.01$) is low.

Performing a repeated measures (rm) ANOVA with time (pre, post) and valence (pos, neg) as within-subject factors and group as between-subjects factor revealed no significant time effect ($F(1,30) = 0.221$, $p = 0.642$) but a significant valence effect ($F(1,30) = 17.992$, $p < 0.001$) as well as a significant time*valence interaction ($F(1,30) = 6.918$, $p = 0.013$). However, no significant group effect ($F(1,30) = 0.171$, $p = 0.682$) as well as no significant interactions with the factor group emerged ($p > 0.335$ in all cases). Applying post-hoc analyses on the significant interaction demonstrated a significant increase in negative mood (post > pre, $p = 0.043$) and a significant decrease in positive mood (pre > post, $p = 0.043$). In general, positive ratings were significantly higher than negative ratings (pre: $p < 0.001$; post: $p = 0.045$).

From the results of the ESR questions (Tables 4 and 5), we learned that the control group experienced an increase in disgust and surprise compared to the experiment group. Also, both groups show an increase in self-reported stress but the standard deviation for the experiment group also increased which implicates that at least some participants experienced less stress. The repeated measures (rm) ANOVA shows a significant condition

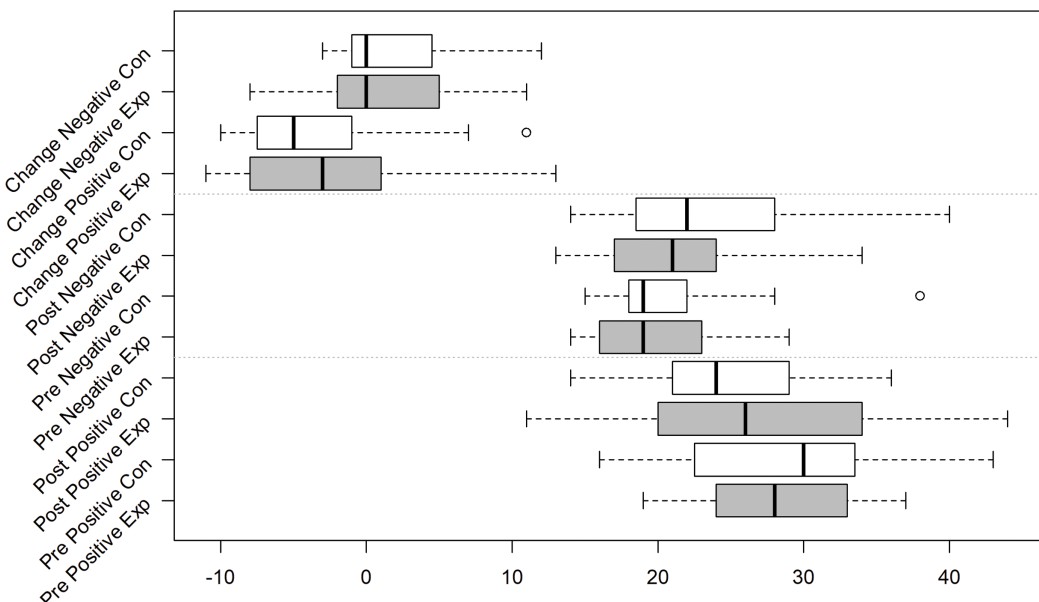

**Figure 5 PANAS Pos(itive) and Neg(ative) Mood Indicators Exp(erimental) Group (Sample size = 10) and Con(trol) Group (Sample size = 11).**

**Table 4 Pre and post-experiment task values for ESR factors, experiment group, study 2, sample size = 9.**

|  | Anger | Disgust | Happiness | Sadness | Surprise | Fear | Stress |
|---|---|---|---|---|---|---|---|
| ESR pre experiment group |  |  |  |  |  |  |  |
| Mean | 1.11 | 1.22 | 2.44 | 1.22 | 2.67 | 1.44 | 2.33 |
| Standard deviation | 0.33 | 0.44 | 0.53 | 0.67 | 1.32 | 0.53 | 0.71 |
| Median | 1.00 | 1.00 | 2.00 | 1.00 | 3.00 | 1.00 | 2.00 |
| ESR post experiment group |  |  |  |  |  |  |  |
| Mean | 1.78 | 1.22 | 2.22 | 1.33 | 2.67 | 1.44 | 2.56 |
| Standard deviation | 1.09 | 0.67 | 1.2 | 0.71 | 1.32 | 0.73 | 1.01 |
| Median | 1 | 1 | 2 | 1 | 3 | 1 | 2 |

effect ($F(6,180) = 25.359$, $p < 0.001$), a trend for a time effect ($F(1,6) = 3.778$, $p = 0.061$), with higher values post than pre, and no significant group effect ($F(1,30) = 0.239$, $p = 0.629$) emerged. Moreover, a significant interaction of condition*time ($F(6,180) = 3.532$, $p = 0.007$) occurred, while no other interaction reached significant ($p > 0.298$ in all cases). Post-hoc tests disentangling the significant interaction revealed a significant increase in anger ratings (post > pre, $p = 0.001$) and a significant decrease in happiness ratings (pre > post, $p = 0.012$). All other comparisons remained non significant ($p > 0.118$).

The cognitive load (see Figs. 6 and 7, missing samples did not finish both *N*-Backs, see "Pilot Study 1" for description of layout) again shows the same effect as seen in the pilot

**Table 5 Pre and post-experiment task values for ESR factors, control group, study 2, sample size = 15.**

|  | Anger | Disgust | Happiness | Sadness | Surprise | Fear | Stress |
|---|---|---|---|---|---|---|---|
| ESR pre control group |  |  |  |  |  |  |  |
| Mean | 1.53 | 1.13 | 2.8 | 1.2 | 2.3 | 1.53 | 2.47 |
| Standard deviation | 0.64 | 0.35 | 0.94 | 0.56 | 0.98 | 1.06 | 0.99 |
| Median | 1 | 1 | 3 | 2 | 2 | 1 | 3 |
| ESR post control group |  |  |  |  |  |  |  |
| Mean | 2.2 | 1.53 | 2.53 | 1.47 | 2.47 | 1.47 | 2.73 |
| Standard deviation | 0.94 | 10.6 | 0.74 | 0.74 | 1.19 | 0.64 | 0.96 |
| Median | 2 | 1 | 3 | 1 | 2 | 1 | 3 |

**NBack Correctness Results**

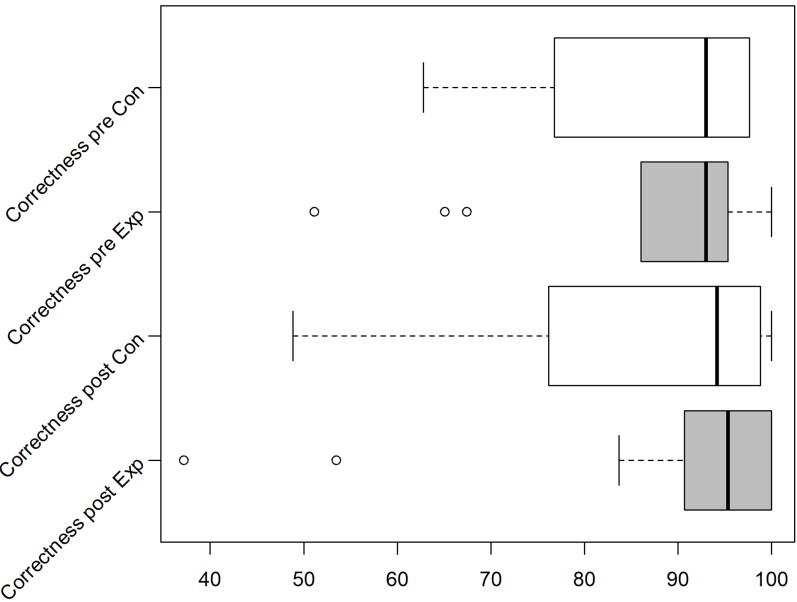

**Figure 6** *N*-Back correctness for the Exp(eriment) Group (Sample size=9) and Con(trol) Group (Sample size=15).

(correctness (mean increased by 1.5% points) and response time improvement (median decreased by about 125 ms) for experiment group vs. correctness decrease (mean by 1% point) for control group) but still is not statistically significant. The Wilcox test does not reveal a statistical relevance ($p = 0.3217$, $\alpha = 0.05$). The slight reduction in response time and increase in correctness for both groups originates most likely from a learning effect.

Performing the rmANOVA with time as within-subject factor and group as between-subjects factor showed a significant time effect ($F(1,24) = 5.389$, $p = 0.029$), no group effect ($F(1,24) = 3.026$, $p = 0.095$) but a significant time*group interaction ($F(1,24) = 6.669$, $p = 0.016$). Disentangling the significant time*group interaction revealed a significant

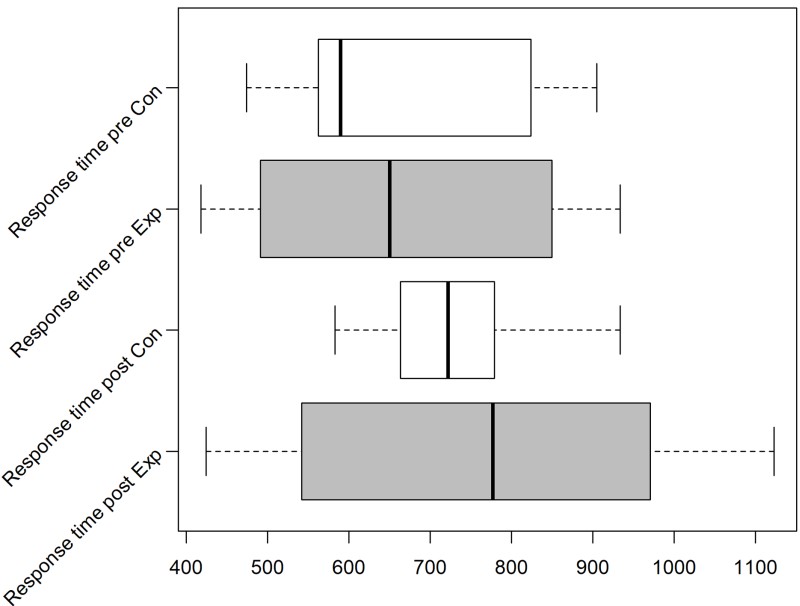

**NBack Response Time Results**

**Figure 7** *N*-Back reaction time for the Exp(eriment) Group (Sample size = 14) and Con(trol) Group (Sample size = 12).

**Table 6 Cortisol and alpha-amylase scores, Study 2.**

|  | Experimental | Control | Experimental | Control | Experimental | Control |
|---|---|---|---|---|---|---|
|  | Offset cortisol (pg/ml) | | Peak cortisol (pg/ml) | | Final cortisol (pg/ml) | |
| Mean | 6.13 | 7 | 4.55 | 5 | 3.83 | 4 |
| Deviation | 3.36 | 2.65 | 2.53 | 2.39 | 2.31 | 2.68 |
| Median | 5.7 | 6.85 | 3.44 | 4.33 | 2.85 | 2.95 |
| Sample Size | 15 | 14 | 15 | 14 | 15 | 14 |
|  | Offset alphaamylase (U/l) | | Peak alphaamylase (U/l) | | Final alphaamylase (U/l) | |
| Mean | 62.99 | 55 | 84.24 | 75 | 94.12 | 91 |
| Deviation | 16.51 | 13.11 | 13.22 | 25.37 | 23.42 | 23.5 |
| Median | 62.4 | 53.85 | 87.4 | 67.75 | 100.4 | 93 |
| Sample Size | 15 | 14 | 15 | 14 | 15 | 14 |

group difference before the intervention (pre: $p = 0.022$), with better performance in the control group. After the intervention (post), no significant group difference emerged ($p = 0.625$). While the experimental group did not show a change in performance (pre vs. post, $p = 0.517$), the control group showed a significant decrease (pre vs. post, $p = 0.041$).

The median of the cortisol measurements (see Table 6) for the control group are slightly higher but the alphaamylase values are lower. However, this test shows no statistical significance (Willcoxon *U* for α = 0.05 for cortisol (C1 $p = 0.35$, C2 $p = 0.4$, C3 $p = 0.68$), Cliff's delta for cortisol (C1:0.21, C2:0.19, C3:0.10), *t*-Test for α = 0.05 for alpha amylase

(A1 $p = 0.14$, A2 $p = 0.25$, A3 $p = 0.07$), Cliff's delta for alpha amylase (A1:−0.31, A2:−0.31, A3:−0.15)). The raw data of the hormone levels can be found in the appendix in Tables A3 and A4.

The rmANOVA with time and group revealed a significant time effect ($F(2,54) = 21.451$, $p < 0.001$), indicating a significant decrease from t1 to t3 (t1 vs. t2: $p < 0.001$; t2 vs. t3: $p = 0.012$; t1 vs. t3: $p < 0.001$), but no group effect ($F(1,27) = 0.178$, $p = 0.676$) nor group*time interaction ($F(2,54) = 0.163$, $p = 0.810$).

Similar to the cortisol analysis, the rmANOVA for the Alphaamylase indicated a significant time effect ($F(2,54) = 34.361$, $p < 0.001$), demonstrating a significant increase from t1 to t3 (t1 vs. t2: $p < 0.001$, t2 vs. t3: $p = 0.003$; t1 vs. t3: $p < 0.001$), but no significant group effect ($F(1,27) = 1.525$, $p = 0.227$) nor a significant group*time interaction ($F(2,54) = 0.262$, $p = 0.771$). This means that the experiment group had a lower chemical stress reaction but a higher need for chemical energy. This might be an indicator that the experiment group reached an energy consuming coping strategy for the given problem sooner. The statistical power increases which indicates, that a larger group or a greater stress induction could show a more prominent effect.

Still, the overall design proved to be feasible and changes made compared to the pilot have shortened the time needed to analyze the generated data.

## Limitations

Based on lessons learned (and the useful suggestions of two anonymous reviewers), we summarize potential limitations that threat the validity of results, should our design be adopted.

## Unreported medical conditions

Participants might refrain to reveal severe medical conditions, like Addison's disease or Cushing's syndrome, which influence the levels of hormones that are measured in our design, but we also believe that such issues have negligible impact on the results of studies from our design. First of all, conditions with severe impact on the measurements are rare, as, for example, Addison affects about 0.9 to 1.4 per 10,000 people in the developed world (*Neto & De Carvalho, 2014*) and Cushing's syndrome is even rarer (*Lindholm et al., 2001*). Also, values arising from pathologically changed hormone values should be visible as outlier in the data.

## Stress induction

The stress induced to participants has to be large enough to be significant and distinguishable from the (quite low) stress induced by the measurement instruments (*N*-Back, saliva samples and questionnaires). We have tried to balance stress induction vs. quality of measurement with our selected tools. Our collected data has shown that our stress induction effect has been too low. To plan the stress induction accordingly, we advise to take a closer look at the different way stress can be induced in psychological research (e.g., *Kirschbaum (2015)* as an example of socially induced stress or *Kang & Fox (2000)* as an example of cognitive stress induction).

### Learning effects of participants

Some stress measurement tools (e.g., the *N*-Back test) base their results on memory and reaction time of the participants. To keep the effect of learning at a minimum the tests should be randomized to the best possible extent. In our case we used two different versions on the *N*-Back (see "Measurement Tools").

### Reuse of the stress task under research

If a task is being reused, participants might give away information about the task to other future participants. The information flow should be restricted if possible, as uncertainty is part of the stress generation (see "Definition of Stress"). While in our case the task was reused, our task consisted of several many subtasks with no clear correct answer: communication would not have done harm. It was also our design to keep participants uncertain about their performance on the task, as well as the task itself.

## LESSONS LEARNED

In the following, we report on our experience and the lessons learned in more detail which we believe are valuable for future research attempts building upon our method.

### On effort and monetary costs

The cost per combined measurement (cortisol and alphaamylase) and basic statistical analysis for one saliva sample was about 5 Euros (Swiss Health Care, 2017, http://swisshealthcare.de). This amounts to about 20 Euros per participant for the pilot and about 15 Euros per participant for the second study. Probably this cost can be reduced by arranging a high-volume contract with a laboratory or finding a cheaper provider of this service. Also, analysis of only cortisol reduces the cost per measurement by up to more than 50%.

Sometimes, as in our case, university departments (e.g., medicine, biology, chemistry) already have a contract with a laboratory or might even be able to provide the analysis themselves.

Compared to other equipment for stress measurement, our method lies in the mid section of overall costs. There are cheap heart rate monitors and skin conductance measurement tools but it depends again on the study and the aspects to be tested if they can be used effectively.

Higher priced tools, like EEG (electroencephalography), can deliver other, maybe more precise results but the process is not easily applicable to large groups. Also, those tools need consumables, like electrodes, driving up the costs. Additionally, we need to keep in mind that with the cost for the hormone analysis equipment we are also covering a small part of the analysis of the results as well, as most labs deliver basic statistical calculations (mean, median, standard deviation,....) with the raw measurement data.

The analysis of the results of other tools mentioned will require the help of medical trained personnel; for detailed results rather than just coarse indications, while with the data provided by the lab we can analyze effects with basic statistics.

No process is worth the effort if it does not deliver results. Our case is somewhat inconclusive. We can identify times with higher and lower stress from our results of the

hormone levels. The additional information gathered (e.g., the *N*-Back results) backs up these results. However, we were not able to find a significant difference between the groups in our studies but we strongly suppose that this is due to other problems within these experiments (e.g., too small sample size due to data loss) or that the effect we were trying to observe was too small to be observed with this kind of design. In conclusion we believe that our proposed measurement process can enable non medical or psychological researchers to examine the influence of stress in processes. Our proposed measurements allow for a more detailed analysis, quick and easy applicable even in larger groups. However, our method can only deliver a first glance at stress-related problems. For an in-depth research on stress effects we strongly recommend to seek cooperation with medical or psychological stress scientists.

## Data protection

We tried to gather as few personal data as possible but our proposed method will collect sensitive data beyond the usual demographic data of software engineering studies, that is, medical data. Besides adhering to the local laws on data protection, we believe that extra care should be taken when gathering, storing, and processing these data.

Before the pilot, we had an extensive discussion with the data protection agency, a German federal agency in charge of enforcing and consulting on data protection laws. We decided to remove the personalization of the data points (pseudo-anonymised data). We talk about pseudo-anonymised data as the recent European data protection law especially has this term within its text in contrast to the old law which defined a term of "anonymised data". Pseudo-anonymisation is reached when no relation can be easily established to the personal data (e.g., names, gender…) in contrast to full anonymisation, where the relation to personal data can be reestablished under no circumstance. In our case, we would only reestablish the connection to personal data if we were able to access to the data of all university students and, even then, only with a tiny probability. In other words, we cannot reestablish the connection.

It is even more important to supply a written statement that explains what data will be gathered, for what purpose and the right to revoke the agreement and, also, the permission to use the gathered data at any time. With his or her signature, each participant should agree to these terms beforehand. This also shows the participants that their personal data will be handled securely and fairly.

## Effectiveness and ethics of stress induction techniques

The issue of putting human participants under stress despite the potential health risks has to be addressed. We believe that the risk of permanent problems as a result of an experiment as described here is highly unlikely. The stress created is only temporary and is not more severe than day-to-day stress peaks. Still, it might be desirable to screen potential participants for preexisting issues which can amplify the negative effects of stress (e.g., a mental disorder). It is also necessary to fully inform the participants about the stress parts of the experiment beforehand if possible. Additionally, we advise to contact an ethics committee, if available, especially if drastic changes to the stress induction are made.

Still, to our experience a formal investigation by an ethics committee is not necessary for this kind of study.

Additionally, we believe that there is a similarity with the issues in controlled experiments observing affect (moods, emotions). As summarized by *Graziotin (2016)*, several studies have doubted the effectiveness of short mood-induction techniques for psychological experiments, where participants' affect is manipulated through several techniques, for example, watching a sad movie, and effective long-term induction techniques might raise several ethical concerns, for example, as with the Facebook emotion contagion experiment (*Shaw, 2016*). Seeing how difficult it has been for us to manipulate stress when employing a robust methodology, we wonder whether the same mechanisms occur for stress induction technique, and if we should rather perform in situ studies. However, this reasoning is speculation at this point. Future studies should address the question of whether stress induction techniques are ineffective for controlled experiments.

## CONCLUSION

In this work, we provided a brief introduction to stress theories and the effects of short-term and long-term exposure on mind and body. We explained how the world of software development is also saturated by stress-inducing events. We discussed how stress can be measured and proposed an efficient way to enable software engineering research to investigate the effects of stress on different processes including software engineering applications. We used our proposed measurement technique in two experiments rendering the approach as feasible and applicable to research on larger groups in software engineering. With this, we hope to enable and make the transfer of medical and psychological methods and knowledge to software engineering easier.

# APPENDIX

## Appendix of raw data
### *Raw Data of Cortisol and Alpha-Amylase Scores*

**Table A1** Results of the measurement of cortisol (picogramm per milliliter) in the gathered saliva samples of the first experiment.

|  | C(S1) (pg/ml) | | C(S2) (pg/ml) | | C(S3) (pg/ml) | | C(S4) (pg/ml) | |
| --- | --- | --- | --- | --- | --- | --- | --- | --- |
|  | Cont. | Exp. | Cont. | Exp. | Cont. | Exp. | Cont. | Exp. |
|  | 3.86 | 2.73 | 4.79 | 3.18 | 3.06 | 3.28 | 3.22 | 5.39 |
|  | 11.4 | 9.82 | 8.22 | 9.72 | 7.57 | 6.77 | 5.86 | 6.32 |
|  | 6.2 | 7.47 | 5.49 | 4.71 | 8.3 | 3.64 | 9.71 | 0.319 |
|  | 7.08 | 5.21 | 8.7 | 4.82 | 5.81 | 10.7 | 5.34 | 6.6 |
|  | 8.67 | 4.74 | 12.6 | 8.5 | 12.5 | 0.508 | 5.42 | 13.6 |
|  | 9.4 | 4.15 | 6.15 | 4.24 | 5.1 | 5.49 | 4.49 | 3.33 |
|  | 9.79 | 3.52 | 8.71 | 13.8 | 7.27 | 15.6 | 0.847 | 6.41 |
|  | 6.06 | 14.9 | 7.66 | 12.4 | 7.4 | 11.4 | 2.58 | 2.02 |
|  | 14.8 | 13.8 | 16.7 | 8.74 | 24 | 8.08 | 4.12 | 2.41 |
|  | 17.8 | 3.08 | 12.7 | 4.23 | 10.8 | 2.32 | 2.79 | 9.22 |
|  | 2.93 | 3.78 | 4.74 | 1.44 | 4.46 | 6.13 | 9.95 | 3.59 |
|  | 7.32 | 2.65 | 1.78 | 1.87 | 5.85 | 3.82 | 1.77 | – |
|  | 7.79 | 1.68 | 16.5 | 0.726 | 4.59 | 11.2 | 4.2 | – |
|  | 10.3 | 7.04 | 7.08 | 3.93 | 4.43 | 5.77 | 0.797 | – |
|  | 2.22 | 0.349 | 2.91 | 0.432 | 2.96 | 5.63 | 6.42 | – |
|  | 6.52 | 5.73 | 4.68 | 5.72 | 3.86 | – | – | – |
|  | 3.64 | 13.5 | 5.58 | 7.25 | 5.3 | – | – | – |
|  | 4.62 | – | 0.579 | – | 0.499 | – | – | – |
|  | 8.26 | – | 6.19 | – | 11.4 | – | – | – |
|  | 8.09 | – | 17.7 | – | – | – | – | – |
|  | 0.584 | – | 0.46 | – | – | – | – | – |
|  | 0.585 | – | 0.66 | – | – | – | – | – |
| Median: | 7.2 | 4.74 | 6.17 | 4.71 | 5.81 | 5.77 | 4.2 | 5.39 |
| Average: | 7.18 | 6.13 | 7.3 | 5.63 | 7.11 | 6.69 | 4.5 | 5.38 |
| *t*-Test: | $p = 0.2919$ | | $p = 0.3134$ | | $p = 0.5417$ | | $p = 0.6098$ | |
| Cliff's Delta | −0.068 | | −0.076 | | −0.3 | | −0.564 | |

**Table A2 Results of the measurement of alphaamylase (international unit per liter) in the gathered saliva samples.**

| | A(S1) (U/l) | | A(S2) (U/l) | | A(S3) (U/l) | | A(S4) (U/l) | |
|---|---|---|---|---|---|---|---|---|
| | Cont. | Exp. | Cont. | Exp. | Cont. | Exp. | Cont. | Exp. |
| | 16.47 | 17.92 | 50.24 | 22.08 | 86.67 | 18.11 | 61.39 | 16.43 |
| | 23.54 | 17.51 | 97.27 | 15.75 | 13.14 | 17.89 | 27.81 | 172.56 |
| | 18.14 | 18.46 | 31.67 | 188.33 | 48.07 | 13.61 | 13.84 | 13.55 |
| | 29.55 | 98.08 | 216.33 | 13.35 | 17.89 | 85.85 | 17.97 | 320.17 |
| | 113.03 | 114.94 | 16.32 | 72.8 | 15.29 | 105.69 | 149.19 | 45.89 |
| | 18.76 | 167.67 | 13.48 | 309.29 | 13.53 | 263.35 | 182.35 | 134.5 |
| | 102.16 | 14.71 | 17.49 | 13.6 | 17.78 | 14.72 | 14.36 | 17.19 |
| | 14.09 | 20.8 | 110.31 | 19.58 | 161.42 | 16.4 | 19.88 | 18.76 |
| | 15.94 | 50.49 | 84.76 | 13.49 | 15.29 | 71.71 | 16.59 | 17.65 |
| | 21.07 | 18.11 | 185.61 | 23.14 | 215.78 | 16.59 | 29.12 | – |
| | 16.59 | 25.34 | 45.62 | 17.4 | 19.61 | – | 16.59 | – |
| | 333.76 | 23.44 | 268.52 | 54.3 | 19.9 | – | 44.8 | – |
| | 38.12 | 16.59 | 62.74 | 14.79 | 63.56 | – | 20.39 | – |
| | 16.59 | 16.59 | 16.59 | 22.16 | 16.59 | – | – | – |
| | 13.79 | – | 165.22 | – | 29.42 | – | – | – |
| | 16.59 | – | 18.03 | – | 18.3 | – | – | – |
| | 203.55 | – | 14.13 | – | 13.09 | – | – | – |
| | 22.16 | – | 182.35 | – | 19.88 | – | – | – |
| | 16.59 | – | 195.4 | – | – | – | – | – |
| | – | – | 16.59 | – | – | – | – | – |
| | – | – | 20.39 | – | – | – | – | – |
| Median: | 18.76 | 19.63 | 50.24 | 20.83 | 18.95 | 18 | 20.39 | 18.76 |
| Average: | 55.29 | 44.33 | 87.1 | 57.15 | 44.73 | 62.39 | 47.25 | 84.08 |
| Wilcoxon $U$: | $p = 0.6366$ | | $p = 0.31325$ | | $p = 0.5417$ | | $p = 0.3597$ | |
| Cliff's Delta | −0.421 | | −0.177 | | −0.576 | | −0.608 | |

**Table A3 Results of the measurement of cortisol (picogramm per milliliter) in the gathered saliva samples on the second experiment.**

|  | C(S1) (pg/ml) | | C(S2) (pg/ml) | | C(S3) (pg/ml) | |
| --- | --- | --- | --- | --- | --- | --- |
|  | Cont. | Exp. | Cont. | Exp. | Cont. | Exp. |
|  | 10.25 | 6.22 | 7.67 | 3.58 | 4.81 | 3.22 |
|  | 4.43 | 4.66 | 4.27 | 9.16 | 3.11 | 6.24 |
|  | 3.41 | 10.37 | 1.14 | 5.08 | 1.21 | 4.30 |
|  | 6.53 | 6.34 | 3.21 | 3.78 | 2.38 | 4.19 |
|  | 7.16 | 4.24 | 3.52 | 3.14 | 2.51 | 2.07 |
|  | 9.22 | 5.72 | 6.37 | 2.70 | 11.90 | 1.89 |
|  | 3.29 | 2.84 | 4.39 | 2.35 | 2.66 | 1.87 |
|  | 4.93 | 6.28 | 2.81 | 6.97 | 2.74 | 8.91 |
|  | 8.45 | 4.25 | 5.25 | 3.22 | 2.94 | 2.83 |
|  | 10.30 | 6.06 | 8.99 | 4.81 | 5.57 | 4.02 |
|  | 9.48 | 5.70 | 9.31 | 3.25 | 6.72 | 2.01 |
|  | 4.91 | 16.40 | 3.89 | 10.90 | 2.84 | 8.47 |
|  | 3.33 | 4.22 | 3.16 | 2.67 | 2.95 | 2.11 |
|  | 8.38 | 5.63 | 5.13 | 3.24 | 3.36 | 2.51 |
|  | – | 2.96 | – | 3.44 | – | 2.85 |
| Average: | 6.72 | 6.13 | 4.94 | 4.55 | 3.98 | 3.83 |
| Median: | 6.85 | 5.70 | 4.33 | 3.44 | 2.95 | 2.85 |
| Wilcoxon $U$: | $p = 0.3536$ | | $p = 0.40$ | | $p = 0.683$ | |
| Cliff's Delta: | 0.2095 | | 0.1905 | | 0.0952 | |

**Table A4 Results of the measurement of alphaamylase (international unit per liter) in the gathered saliva samples for the second experiment.**

|  | A(S1) (U/l) | | A(S2) (U/l) | | A(S3) (U/l) | |
| --- | --- | --- | --- | --- | --- | --- |
|  | Cont. | Exp. | Cont. | Exp. | Cont. | Exp. |
|  | 47.1 | 60 | 54.8 | 89.3 | 85.4 | 90.6 |
|  | 46.2 | 78.1 | 50.9 | 55.7 | 88.8 | 62.6 |
|  | 65.6 | 53 | 136.3 | 87.4 | 139.4 | 84.2 |
|  | 67.7 | 61.7 | 82.5 | 92.5 | 99.1 | 85.8 |
|  | 53.3 | 77.3 | 90.2 | 84.9 | 97.2 | 101.3 |
|  | 35.6 | 41.1 | 102.66 | 98.4 | 49.55 | 104 |
|  | 76.4 | 84.19 | 59.15 | 89.8 | 110.31 | 129.9 |
|  | 55.8 | 66.3 | 92.4 | 64.9 | 98.8 | 34.9 |
|  | 40.09 | 49.7 | 65.8 | 92.7 | 93.1 | 105.25 |
|  | 38.8 | 63.5 | 43.5 | 78.8 | 51.3 | 87.1 |
|  | 49.9 | 42.99 | 52.1 | 96.1 | 72.2 | 100.4 |
|  | 76.6 | 35.02 | 90.4 | 69.1 | 111.8 | 83.11 |
|  | 57.9 | 88.1 | 62.8 | 104.5 | 92.9 | 117.6 |
|  | 54.4 | 81.5 | 69.7 | 80.8 | 77.7 | 114.3 |
|  | – | 62.40 | – | 78.7 | – | 110.80 |
| Average: | 54.67 | 62.99 | 75.23 | 84.24 | 90.54 | 94.12 |
| Median: | 53.85 | 62.4 | 67.75 | 87.4 | 93 | 100.4 |
| $t$-Test: | $p = 0.1434$ | | $p = 0.2497$ | | $p = 0.0684$ | |
| Cliff's Delta: | −0.3143 | | −0.3143 | | −0.1524 | |

## Questionnaires
### Socio-demographic questions
PANAS and ESR (in German translation as used by *Schneider et al. (1994)*)
Self-efficacy questions

## ACKNOWLEDGEMENTS

We thank our participants for taking part in our study. We are grateful for the feedback provided by two anonymous reviewers and the editor. Our thanks to Katharina Plett for the professional proofreading of this article.

### Funding
The authors received no funding for this work.

### Competing Interests
Stefan Wagner is an Academic Editor for PeerJ.

### Author Contributions
- Jan-Peter Ostberg conceived and designed the experiments, performed the experiments, analyzed the data, performed the computation work, prepared figures and/or tables, authored or reviewed drafts of the paper, and approved the final draft.
- Daniel Graziotin analyzed the data, authored or reviewed drafts of the paper, and approved the final draft.
- Stefan Wagner analyzed the data, authored or reviewed drafts of the paper, and approved the final draft.
- Birgit Derntl conceived and designed the experiments, analyzed the data, authored or reviewed drafts of the paper, and approved the final draft.

### Data Availability
    All the data is available in the Appendices.

### Supplemental Information
Supplemental information for this article can be found online at http://dx.doi.org/10.7717/peerj-cs.286#supplemental-information.

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
