# Peer review of "A methodology for psycho-biological assessment of stress in software engineering"

_PeerJ Computer Science, doi:10.7717/peerj-cs.286_

## Round 0.1 · original submission · Major Revisions

This paper proposes a protocol to measure stress in a software engineering context. The paper offers a thorough analysis of related work on stress in a software engineering context. It then provides a summary of existing work on defining stress and its effects, and offers an overview of measuring stress using psychological measurements or biological markers.

Based on this, the paper proposes a protocol for measuring stress in software engineering experiments. The protocol combines PANAS, the Positive And Negative Affect Scale, and ESR, the Emotional Self Rating scale for the psychological and emotional stress measurements. The protocol is used in two small pilot projects, demonstrating the feasibility of the protocol (although no significant findings were drawn from the two pilots).

The ethics of stress induction is discussed in 7.3, and briefly in 6.1. The paper comes with two German letters from the ethics committee of the University of Tübingen, which appear to say that consultation of the ethics committee is not compulsory for this type of study. It would maybe be helpful to add a translation of the two letters to the dossier (as well as of Appendix A.2). In 7.3 it would be good to be explicit about the need for Ethical Review Board approval for this type of study.

The reviewers see many positive aspects of the paper, but also identify a number of constraints that must be addressed:

1. Reviewer 1 raises several questions (under the header Experimental Design) about sections 5 and 6. Please adjust these sections so that these questions are answered in the text

2. Reviewer 1 raises concerns (under validity of the findings) about the reporting of the statistical significance. The reviewer offers various suggestions to address these concerns.

3. Reviewer 2 raises concerns about the robustness claims (under experimental design) and asks for clarifications

4. Reviewer 2 indicates that, while the examples are drawn from software engineering, the proposed protocol is not very software engineering specific. This is an issue to discuss, on which the paper needs to offer a clear point of view (is this good / by design? Is this a problem that should be mitigated?)

5. Reviewer 2 raises a concern about threats to validity. In fact, I would suggest that this paper could benefit from a generic "threats to validity" section that would inform anyone following this protocol about the threats that they need to be aware of.

Reviewer 1 ·

Basic reporting

- Overall, the paper has been written clearly but there are too many typos, misspellings, punctuation problems and inconsistencies that need to be fixed. A few examples:
Line 13: exposere -> exposure
Line 28: research on of stress -> research on stress
Line 36: Wordl … realiced -> World … realised
Line 192: Open double quotation for “unchangeable”, both are close double quotations.
Lines 194-195: This is becomes … -> This becomes …
Line 196: missing space after semicolon and again problem with double quotations.
...
Line 516: Be consistent with the usage of “alphaamylase” or its other form.
...

- The paper provides sufficient background and it is well structured.

- Figures must be revised to have better quality, especially when printed. All the figures (except figure 1) need better labeling and information support in the paper's main text. For example, it would be nice if you add more information in the text about the range of scores in PANAS. Or what is Div. Responsetime in Figure 4 and why does it have a negative median?
Figure 7 (a): It seems that the sample size is a typo.

- The orientation of Tables and their captions need to be fixed.

Experimental design

The Methodology and case studies are nicely designed and aligned with the aim of the study. The split of each stage of the experiment is sound. However, based on the information that the authors provide as the background of the study, important questions come to mind:

- Section 5.1 states that there is a possibility that stress reactions be influenced by medication or pre-existing mental conditions. To what extent can they tell what portion of the stress reactions are due to secondary reasons rather than the task itself? Which one has a larger effect: the task, medications or mental conditions?
(I also could not find any notes in this paper regarding your participants. It would be nice to report if none of your participants had medical or mental conditions.)

- Section 5.2 points to the limitations of using hormone tests, as the hormone system reacts in the range of minutes to hours and cortisol have a daily cycle. In psychological studies, most of the time there are not strict participation requirements, but it can easily become complicated to find software engineers who accept to participate in such a setting.
As the authors are trying to propose a standard framework for the software engineering domain, do they think researchers can follow such a framework?
Finding participants is a challenge itself. Is there a possibility that these conditions make software engineers refuse to participate?

- What if a participant has a health condition that they are not aware of or they don’t want to share with you? Like Addison's disease or Cushing's syndrome which directly affects cortisol?

- Section 6.1 states that in the pilot study, you split the task into two parts of 25 and 20 minutes. In case a participant finishes the first part in less than 10 minutes, can we expect that the saliva test works (as the system responds in 15-20 minutes)?

- As a side note, section 6.1 does not clearly state which samples are missing. Are the same samples missing for NBack correctness in Figures 3 and 4? Are there any overlaps in missing data from PANAS Indicators and NBack correctness?

Validity of the findings

I understand and appreciate the time and energy that the authors invested in conducting this study. Their study design is thoughtful. Yet, analysis and conclusions are not supported. In its current shape, the methodology does not show enough evidence to be capable of being a guiding framework for future studies.

- First case study (pilot study):
How reliable is 4 and 6 samples to draw any conclusions and move to the next case study?
Authors admit that they did not find anything significant in the pilot study but they state that the study is feasible. Is the feasibility of a study enough to move it to the next stage? If they wanted to just check the feasibility, why didn’t they recruit, let’s say, 5 students? What was the point to recruit 46 people and then conclude that the study is feasible?
In my opinion, the huge amount of missing data in the pilot phase, which led to non-significant results, discredits the paper’s contribution. Hence, my suggestion is to revise the paper’s abstract and do not emphasize the pilot study as a case study and a contribution.

- Lines 522-529: This sample size does not grant enough evidence to even run significance tests. Hence, discussions on Lines 524 and 525 seems pointless.

- Lines 541-543: Any significant stress results? Lines 522 and 523 stated that there was not enough data for the n-back test and results may get meaningful with a larger dataset, how come non-significant results led to ignoring baseline and offset but not other stages of the study?

- Line 557: Master students have been recruited for your pilot study and Bachelor students for the second study. Does the task fit both Bachelor and Master students similarly?

- Lines 577-578: Reporting these findings based on "mean" while the standard deviation is quite large and the median is constant does not seem appropriate.

- Section 7.1 does not seem convincing enough. The cost of other equipment, such as eye-trackers or heart rate monitors, is usually a one-time cost and they can be reused for several studies. Moreover, these listed costs can vary from country to country and the laboratories may not be accessible to many researchers.

Additional comments

The following are some suggestions that can enhance the clarity of the paper.

Line 389: Cite a number of such studies here.
Lines 438-439: What is the exact basis of this advice?
Lines 445-446: Cite the most effective papers that shaped your methodology here.
Line 469: In line 460 you stated that the endocrine system responds in 15-20 minutes after the stressor onset. Why in line 469 you suggest 40-120 minutes for longer studies?
Lines 495-497: It is not clear what was the intention by stating mean, median and standard deviation. Boxplots are a standardized way of displaying five number summary (minimum, first quartile, median, third quartile, and maximum). Did you want to mention the content of Table 1?
Line 498: Line 485 reported a total of 46 participants for the pilot study but in line 498 the total number adds up to 50. Which one is correct?
Lines 510-511: What are those 22 and 17 measurements? It would be nice to cite a reference for it. Why are they not equal for the experiment and control group? What do those measurements tell us? What should I learn from Table 2?
Lines 536-540: Are there any other reasons rather than reducing time and money? If so, please clarify.
Lines 557-561: Add an explanation of why there are missing samples in Figures 5 and 6.

Reviewer 2 ·

Basic reporting

The use of the English language is mostly clear and professional with few exceptions which could be easily fixed with a further round of proofreading.
Regarding style, some sentences can be simplified (e.g., Section 1, lines 31–37).

The paper reports enough background to be understood by a technical audience in the field of software engineering not familiar with physiology and psychometrics. An ad-hoc section (Section 3) is devoted to cover the basic required concepts.
Relevant prior literature for software engineering is reported in Section 1 and Section 2. However, I find the second part of Section 1 (starting from line 66) a bit disconnected from the rest. Given its contents, it feels it should belong to Section 2. I understand these paragraphs are necessary for motivating the need for interdisciplinary research, but the gap should be made more clear (at the moment, these paragraphs only enumerate previous interdisciplinary work).

Although it follows a customary order for SE papers, the structure of the article can be improved by moving Section 2 after Section 6 (i.e., the results of the two studies). This way, previous results on stress can be better contrasted with the results reported in this paper (given that the authors agree that reporting such contrast can be of interest). Similarly, factors of interest—e.g., the cost for running a study reported in Section 7—can be easily compared by, for example, formulating a hypothetical scenario in which physiological techniques are used in previous work.

Boxplots comparing different treatments (i.e., groups) should be reported side by side, for a better comparison, in a single plot.
The boxplots axes labels are currently unreadable.
Tables can be arranged so that they do not need to be reported in landscape layout.
The information regarding the results of the statistical test should be reported in the text, rather than in Table 2 caption.
In my opinion, the data is made available to the best extent possible (Appendix A).

The paper, in its current form, is self-contained. The study reported (including a pilot) and the references cited contribute to its relevance with respect to its goals.

Experimental design

The research is relevant to the field of software engineering. However, research questions are not formally defined. A gap is identified in Section 1 but (as commented in the section above) it should be made more clear based on the cited references.
Contributions are clearly stated. The first one is supported clearly in the paper (i.e., Section 4). The second is phrased as "a robust methodology to detect and measure stress in controlled experiments that is tailored to software engineering research."
I have two main objections regarding the latter, i) I understand a methodology to be robust if it consistently (e.g., under several runs) detects what it is supposed to detect. This is not the case according to the (only two) studies reported; and, ii) it is not clear how the methodology (which really is an experimental design + a methodology for stress detection) is specific for software engineering experiments. One could just replace the experimental tasks with tasks specific to another discipline.

The study design seems solid and results are reported for both studies. The authors also reflect on the activities undertaken with respect to ethical conduct and data protection.

The methodology is explained to a good level of details (including figures and annexes) which would aid replication. Moreover, the authors reflect on the cost of replicating the methodology in terms of money, effort, and required skills.
However, the evaluation method should be more precise. For example, "The methodology has been supported by two case studies implemented as controlled experiments..." these are actually controlled experiments (base + replication) and not case studies. I suggest to report them under another lense—i.e., as replications—following the accepted guidelines and reporting standards in software engineering research (see the work of Carver, Vegas, Juristo). Also, the methodology is "evaluated" rather than "supported" by an experiment/case study/etc.

Validity of the findings

The paper does not include a formal "threats to validity" section. However, the authors identify are report some threats in Section 6 and Section 7. I recommend grouping the considerations on the threats to the validity of the experiments in an ad-hoc section (especially in the case the authors will follow my previous recommendation to restructure the paper according to the reporting standard mentioned above).

The paper reports detailed results regarding the experiments as well as interesting lessons learned grounded in their experiences. However, these are targeted to researchers, whereas no implications of practitioners are provided.
Data is reported in the paper in tabular form, and material attached as a supplement to this submission. I believe that, upon acceptance, these materials will be made public.

Additional comments

I am adding minor comments here:
- Please avoid restating the abstract verbatim in the introduction
- The abstract should mention the evaluation methodology and report of a glimpse of the results
- I am not sure the citation style is correct. (Author (Year)) seems odd
- Lines 63–65 "For detecting stress, research in software engineering has focused on machine learning and data mining approaches, wearable technologies, and ad-hoc questionnaires so far (Suni Lopez et al., 2018; Brown et al., 2018; Meier et al., 2018), but we still see some research gaps here." Please, cite the work more precisely (who reported on machine learning, who did so on questionnaires, etc).
- Line 139. "As the difficulty of a task can be a stressor, brain waves are an interesting indicator to measure. However, this kind of study needs specialized equipment and the results are controversial." Why are they controversial? These need to be back up by a solid argument and (possibly) reference to existing evidence.
- Line 159. "While all previous studies contributed to our understanding, there are indications that software engineering has been avoiding using robust and validated methodology for stress detection and psychological issues in general, thus threatening the validity and reliability of studies." This is also a strong claim which should be at least better discussed. Can you report an example of such evidence from Graziotin et al. 2015b?
- Line 161. "Most of these studies use non-standard, ad-hoc, and non psychometrically validated questionnaires to assess the stress reaction, either by self-report or a number of questions aimed to derive the personal stress level." Same as above. Please, indicate more precisely which studies use non-standard questionnaires (with examples).
-This is a wish, and not necessarily a critique. It seems that stress can be related to the workplace (e.g., the work culture) and/or the individual's personality. Perhaps, this can be elaborated a bit more in Section 3.
- Line 478. "Salutogenesis is a well-being theory which we have previously proposed for enhancing the interaction of software developers with their tooling (see Ostberg and Wagner (2016) and Ostberg et al. (2017) for details on Salutogenesis)." This can be explained in a sentence without having the reader look for the reference for the basic definition.
- The last two references (World Health Organization) should be footnotes.

---

## Round 0.2 · Minor Revisions

Thank you for your revision, and apologies for the delay in answering -- present times are a bit too event-ful.

Of the two original reviewers, one is happy without comments about the changes made; The other reviewer indicated they had reached their "quota of reviews of this semester", so declined to review the revision.

I studied your rebuttal and revised paper.

Thank you for your detailed (18 page) rebuttal, and your thoughtful revision of the manuscript. Your paper offers an informed proposal for experimental procedures to measure stress of software engineers, including two case studies to assess the difficulties in applying these procedures. This is an important stepping stone towards more research in the area of the impact of stress in software engineering, and I am happy to recommend acceptance.

In the rebuttal you refer to section 7 as "Threats to Validity", but in the paper its title is "Limitations". I am of the opinion that "Threats to Validity" would be the better title, but leave this to you.

One other detail: Going through the paper I was surprised to see various strong claims that are not substantiated in the paper, and, in fact, are not relevant either for the main result:

- "Software engineering research, regretfully, is a long way from adopting rigorous and validated research artifacts."
- "Given, that much research in the field has misinterpreted, if not ignored, validated methodology and measurement instruments coming from psychology,"

Either provide references with evidence, or omit such sentences. I think it only distracts from your main message on how to measure stress in this paper, and it is better to omit it, even if you could back it up with evidence.

Reviewer 1 ·

Basic reporting

'no comment'

Experimental design

'no comment'

Validity of the findings

'no comment'

---

## Round 0.3 · accepted · Accept

Thanks for taking care of the final comments. I am very happy to see this paper on stress in software engineering being published and recommend acceptance for this paper.